# Nipah virus W protein harnesses nuclear 14-3-3 to inhibit NF-κB-induced proinflammatory response

François Enchéry[1,3], Claire Dumont [1,3], Mathieu Iampietro [1,3], Rodolphe Pelissier [1,3], Noémie Aurine[1], Louis-Marie Bloyet [1], Caroline Carbonnelle[2], Cyrille Mathieu [1], Chloé Journo [1], Denis Gerlier [1,3] & Branka Horvat [1✉]

Nipah virus (NiV) is a highly pathogenic emerging bat-borne *Henipavirus* that has caused numerous outbreaks with public health concerns. It is able to inhibit the host innate immune response. Since the NF-κB pathway plays a crucial role in the innate antiviral response as a major transcriptional regulator of inflammation, we postulated its implication in the still poorly understood NiV immunopathogenesis. We report here that NiV inhibits the canonical NF-κB pathway via its nonstructural W protein. Translocation of the W protein into the nucleus causes nuclear accumulation of the cellular scaffold protein 14-3-3 in both African green monkey and human cells infected by NiV. Excess of 14-3-3 in the nucleus was associated with a reduction of NF-κB p65 subunit phosphorylation and of its nuclear accumulation. Importantly, W-S449A substitution impairs the binding of the W protein to 14-3-3 and the subsequent suppression of NF-κB signaling, thus restoring the production of proinflammatory cytokines. Our data suggest that the W protein increases the steady-state level of 14-3-3 in the nucleus and consequently enhances 14-3-3-mediated negative feedback on the NF-κB pathway. These findings provide a mechanistic model of W-mediated disruption of the host inflammatory response, which could contribute to the high severity of NiV infection.

[1] CIRI, Centre International de Recherche en Infectiologie, Inserm U1111, CNRS, UMR5308, Univ Lyon, Université Claude Bernard Lyon 1, École Normale Supérieure de Lyon, Lyon, France. [2] INSERM- Laboratoire P4 Jean Mérieux, 21 Avenue Tony Garnier, 69365 Lyon, France. [3] These authors contributed equally: François Enchéry, Claire Dumont, Mathieu Iampietro, Rodolphe Pelissier, Denis Gerlier. ✉email: branka.horvat@inserm.fr

Transcription factors of the nuclear factor-κB (NF-κB) family regulate numerous genes involved in the immune response, cell proliferation, differentiation, and death[1,2], and often serve as targets for viral interference[3]. Members of the NF-κB family (p65, p50, c-Rel, RelB, and p52) share a conserved Rel Homology Domain that mediates the formation of hetero- and homodimers (such as the active p65/p50 complex) capable of binding DNA. Normally, p65/p50 heterodimers are retained in the cytoplasm of resting cells due to their association with IκB inhibitors[4]. The NF-κB pathway is triggered via multiple sensors of the innate immunity, including RIG-I-like receptors and Toll-like receptors (TLRs), and activates cellular antiviral programs. Immune sensors can activate the IκB kinase (IKK) complex composed of a scaffolding protein IKKγ (NEMO) and two kinases, IKKα and IKKβ. When activated, these kinases phosphorylate the IκB subunit, which leads to its ubiquitination and subsequent targeting to the proteasome for degradation[4]. Once free from its inhibitor, the p65/p50 complex becomes phosphorylated, gets into a transcriptionally active state, and enters the cell nucleus[5]. There, it binds to a large set of promoters to activate hundreds of different genes, including numerous cytokines[6,7]. Multiple regulators, including 14-3-3 proteins and IκB inhibitors, allow fine-tuning of the NF-κB pathway[4,8–10].

Several pathogenic viruses have been found to target the NF-κB pathway by blocking the nuclear import of p65. For instance, the nucleocapsid protein of Hantaan virus limits p65 nuclear translocation, presumably by interfering with importin-α (also called karyopherin), a nuclear import molecule family responsible for shuttling NF-κB to the nucleus through a putative nuclear localization signal (NLS)[11]. Japanese encephalitis virus NS5 competitively inhibits NF-κB nuclear translocation by binding to importins α2, α3, and α4, likely through two NLSs within the NS5 protein. However, NS5 itself does not accumulate in the nucleus[12]. Herpes simplex virus 1 UL24 binds to p65 and p50 subunits and prevents their nuclear translocation[13]. Finally, Middle East respiratory syndrome-related coronavirus protein 4b appears to outcompete NF-κB for importin α4 binding and translocation into the nucleus[14].

Nipah virus (NiV), a member of the *Henipavirus* genus, is a zoonotic paramyxovirus discovered in 1998 during an encephalitis outbreak among pig farmers in Malaysia[15]. NiV is naturally hosted by *Pteropus* fruit bats, which seem to control NiV infections by a mechanism still poorly understood (reviewed in ref. [16]). Spillovers from bats into humans have been causing near-annual outbreaks of severe encephalitis and acute respiratory disease with up to 90% lethality in Bangladesh and India since 2001[17,18]. NiV is known to preferentially target endothelial cells, neurons, and respiratory tract epithelial cells. The Bangladesh strain of NiV is associated with frequent person-to-person transmission and may possess pandemic potential (reviewed in ref. [19]). Currently, neither approved vaccines nor treatments against human NiV infection exist. Therefore, NiV is classified as a biosafety level 4 pathogen and is considered to be a major biosecurity threat[20]. In 2015, the World Health Organization listed NiV among the top eight emerging pathogens most likely to cause severe outbreaks and placed it on the Blueprint list of priority infectious diseases requiring urgent research and development efforts[21].

The NiV genome comprises six genes coding for the structural proteins N, P, M, F, G, and L (Fig. 1a). In addition to the P protein, the *P* gene encodes for three nonstructural proteins: V, W, and C[22]. Although the C protein is translated from an alternative start codon, the V and W proteins originate from +1 and +2 shifts, respectively, of the open reading frame resulting from the addition of guanosine residues at an editing site of the *P* gene. Thus, the P, V, and W proteins share a common 407-amino-acid-long N-terminal domain (NTD) and differ by their respective C-terminal domains (CTDs) (Fig. 1a). The sequence of the W protein is highly conserved between NiV Malaysia and NiV Bangladesh strains, sharing 100% amino acid sequence identity for W-CTD (Supplementary Fig. 1). V, W, and C proteins are considered as virulence factors that counteract cellular antiviral defenses[23–25]. They prevent the triggering of interferon (IFN) signaling pathways[26,27] and limit type I IFN production[28–30]. In addition, the W protein down-modulates the production of numerous cytokines during NiV infection in vitro, which could limit leukocyte recruitment and increase the severity of respiratory disease in vivo[24]. The W protein is imported into the nucleus via importin α3- and α4-mediated mechanisms. There, it interacts with the TLR3 signaling pathway[28]. However, the detailed molecular mechanisms by which the W protein controls cytokine production are currently unknown.

We report here that NiV infection inhibits the NF-κB pathway via the interaction of the W protein with 14-3-3 cellular proteins. Our data allow constructing a model in which W-S449 binds to 14-3-3, inducing nuclear accumulation of 14-3-3, enhancing NF-κB p65 subunit export into the cytoplasm, and thus downregulating proinflammatory cytokine production. We propose that this molecular mechanism plays a part in NiV immunopathogenesis by counteracting the host inflammatory response and may contribute to the high mortality rate of NiV infection.

## Results

**NiV suppresses the production of proinflammatory cytokines in infected cells and W-CTD plays a role in inhibiting the canonical NF-κB pathway.** As NiV infection is associated with a particular inflammatory response (reviewed in refs. [19,31]), we initially analyzed whether NiV infection in non-human primates could modulate the production of cytokines known to be under the control of NF-κB[6]. African green monkeys (AGMs) are highly susceptible to NiV infection and display disease progression and symptomatology resembling those in humans[32]. Indeed, following infection with wild-type (*wt*) NiV via the intratracheal route, infected monkeys rapidly developed pneumonia with a severe respiratory syndrome and had to be killed within 8–10 days post infection. In infected AGMs, the plasma level of different NF-κB-mediated cytokines, including interleukin (IL)-1β, IL-6, tumor necrosis factor-α (TNFα), macrophage inflammatory protein (MIP)-1α, MIP-1 β, and IL-12/23, remained relatively stable, without any significant increase during the course of severe, eventually lethal infection (Fig. 1b). These cytokines are produced by a variety of cell types and most of them are known to be targeted during NiV infection. By contrast, the levels of NF-κB-mediated IL-2, IFNγ, and perforin, which are primarily produced by T lymphocytes and NK cells[33,34], increased significantly during infection (Fig. 1c), suggesting the establishment of a certain response of these immunocompetent cells. As lymphocytes are not permissive to NiV infection[35], one cannot explain their reaction to the infectious stimulus by intracellular NiV-mediated control of cytokine production. These results suggest that NiV infection could generate a specific cytokine signature, possibly via inhibiting the expression of NF-κB-controlled cytokines in cells productively infected by NiV.

We further explored the effect of NiV infection on NF-κB activity using HEK293T cells, which constitutively express a luciferase reporter gene under the control of a synthetic NF-κB promoter and are permissive to NiV infection. Following viral infection with either a recombinant NiV (rNiV) or rNiV deficient for the CTD of the W protein (rNiV-W$_{\Delta CTD}$)[36], the viral proteins N and P were expressed at similar levels for both

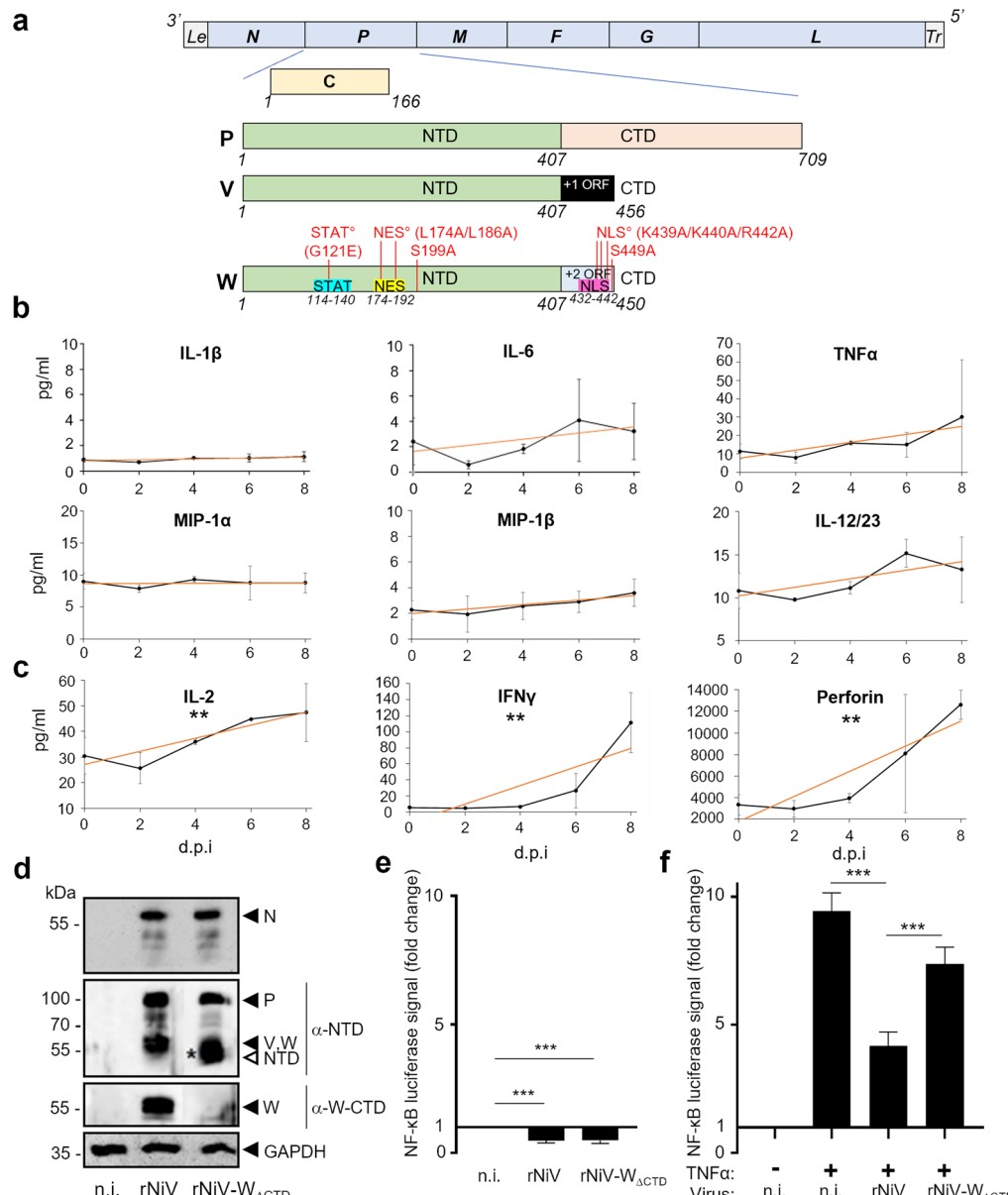

**Fig. 1 W-CTD is necessary for the inhibition of the canonical NF-κB pathway following NiV infection. a** Schematic representation of the NiV genome and the proteins encoded by the P gene: P, C, V, and W proteins. Critical residues of NES, NLS, and STAT-binding sites within the W protein and residue substitutions made in this work are indicated in red. Le, Leader; Tr, Trailer. **b**, **c** Plasma from NiV-infected African green monkeys ($n = 3$) was collected at indicated days post infection (d.p.i.) and analyzed for the presence of NF-κB-controlled cytokines and perforin produced either by a large set of cell types (**b**) or principally by T and NK cells (**c**), using the Milliplex NHP Cytokine assay, with measurements for individual animals done in duplicate. Results are expressed as the mean of concentrations, with error bars representing the SD. Linear regressions were performed (red straight lines) to evaluate the statistical significance of the slopes (**$p < 0.01$). Details of statistical analyses are given in Supplementary Table 1. **d** HEK293T NF-κB-luc cells were infected with rNiV or rNiV-$W_{\Delta CTD}$ at an MOI = 3. At 24 h post infection, cells were lysed and the expression of N protein, P gene-encoded proteins (P, V, W, NTD fragment (common to P, V, and W proteins)), was assessed by western blotting using indicated anti-N, anti-NTD, or anti-W-CTD antibodies. V and W proteins migrate with similar apparent molecular mass and thus cannot be distinguished. The NTD truncated protein (open arrow, *) encoded by rNiV-$W_{\Delta CTD}$ migrates with a lower apparent molecular mass. **e** HEK293T NF-κB-luc cells were infected or not (n.i.) with rNiV or rNiV-$W_{\Delta CTD}$ (MOI = 3) and tested at 24 h.p.i. for NF-κB activity by luminescence quantification. Transfection with *Renilla* luciferase was used for the normalizing of the obtained results. **f** Same conditions as in **e**, with or without additional stimulation with 10 ng/mL of TNFα for 4 h prior to luminescence quantification. The data represent the mean values of at least three independent experiments with each point done in triplicate, presented as mean ± SD. ***$p \leq 0.001$ using one-way ANOVA completed by a Tukey's multiple-comparisons test.

viruses (Fig. 1d and Supplementary Fig. 1b), consistently with the findings of a previous report[37]. As expected, W-CTD was not detected in cells infected with rNiV-$W_{\Delta CTD}$ and replaced by the appearance of the NTD truncated protein (Fig. 1d, white arrow). Although the basal luciferase activity of NF-κB was highly reduced in cells infected with both viruses (Fig. 1e), strong luciferase expression induced by TNFα was significantly inhibited after infection by rNiV and only moderately inhibited after infection by rNiV-$W_{\Delta CTD}$ (Fig. 1f). These results suggested that NiV infection inhibits the TNFα-stimulated NF-κB pathway and that the CTD of the W protein is critical for this suppressive effect.

**W protein inhibits the canonical NF-κB pathway downstream of IKKβ within the cell nucleus and decreases the intranuclear accumulation of the NF-κB p65 subunit**. The strategy used by the W protein to inhibit the NF-κB response was further analyzed using cells transfected with the NF-κB reporter gene and plasmids coding for different W protein variants tagged with FLAG peptide at their N terminus or for a truncated construct where CTD and NTD were replaced by mCherry to give rise to W-NTDm and mW-CTD, respectively. The expression of the W protein significantly inhibited the NF-κB pathway, both in the absence of stimulation and after 4 h stimulation with TNFα or IL-1β (Fig. 2a). This inhibitory effect relies on W-CTD, as it was observed when W-CTD but not W-NTD (common to P and V proteins; Fig. 1a) was expressed within the FLAG-mCherry protein backbone (Fig. 2a, compare mW-CTD with W-NTDm constructs). Thus, W-CTD is necessary and sufficient to inhibit the canonical NF-κB pathway. Correspondingly, the expression of W protein was found to significantly reduce the accumulation of the NF-κB p65 subunit in the nucleus after cell stimulation with IL-1β, as quantified by image cytometry (Fig. 2b, c). The transfection efficiency and cell viability were analyzed for basic conditions (with or without W protein expression and with or without IL-1β stimulation) and were similar between these tested conditions (Supplementary Fig. 2). The activation of the NF-κB pathway by overexpression of IKKβ or TNF receptor-associated factor 6 (TRAF6), an essential adaptor protein in the IL-1β signaling pathway[38], was also inhibited by the W protein (Fig. 2d). In contrast, the NF-κB activity was not reduced by the W protein upon overexpression of p65. Altogether, these results suggest that the NF-κB inhibitory effect of the W protein occurs downstream of IKKβ, but upstream of p65, as it cannot counteract stimulation by overexpressed p65.

**Nuclear localization of W protein is required for its activity**. Nuclear localization of the W protein is necessary for the inhibition of TLR3 signaling[28,39] and may have an important role in its additional activities. We assessed whether the abrogation of the nuclear export signal (NES), the NLS or both could affect the NF-κB pathway using the L174A-L186A (W-NES[0]) and K430A-K440A-R442A (W-NLS[0]) mutants[26,28] (Fig. 1a). When W-NES[0], predominantly nuclear (Supplementary Fig. 3a) due to the lack of nuclear export[26], was expressed in HeLa cells, the NF-κB inhibitory effect was preserved (Fig. 2e). Conversely, W-NLS[0], mainly cytosolic (Supplementary Fig. 3a) due to the impairment of import[28], was unable to inhibit the NF-κB signaling activated by IL-1β (Fig. 2e). Finally, W-NES[0]NLS[0], which was widely distributed both in the nucleus and the cytosol (Supplementary Fig. 3a), was proved inactive (Fig. 2e).

Residues 114–120 of the NTD bind STAT1 and STAT4[40] (Fig. 1a), and the G121E W protein variant (W-STAT[0]) is unable to interact with the STAT1 protein[27]. This variant retained the ability to inhibit the NF-κB response (Fig. 2e), as expected, following our finding that the W-NTD is dispensable for this inhibition (Fig. 2a). In conclusion, the W protein acts downstream of IKKβ by decreasing the amount of the intranuclear NF-κB p65 subunit. This effect relies on the ability of the W protein to be imported into the nucleus.

**W protein interaction with endogenous 14-3-3 is dependent on W-CTD in both transfected and NiV-infected cells, and the interaction relies on S449 residue**. We next screened for cellular proteins interacting specifically with W-CTD by mass spectrometry (MS) analysis. HeLa cells were transfected with plasmids coding for FLAG-tagged V and W proteins. NiV proteins and their interactors were eluted using magnetic beads bound to anti-FLAG antibodies and analyzed by MS. Proteins specifically coimmunoprecipitating with the W-CTD were obtained by subtracting the list of all V interactors from the list of W protein interactors, as W and V proteins share the NTD region but have distinct CTDs (Fig. 1a). Three of the identified cellular proteins (importin subunit α4, CDC5L, and 14-3-3θ) had been reported previously[28,39,41,42]. As members of the 14-3-3 protein family are known to act as crucial regulators of transcriptional pathways[43], including the NF-κB pathway[10], we focused our investigations on the possible role of 14-3-3.

The selective interaction of the W and mW-CTD proteins with endogenous 14-3-3 and the inability of the V and W-NTDm proteins to immunoprecipitate (IP) 14-3-3 was further confirmed by coimmunoprecipitation by anti-FLAG antibody followed by western blot analysis (Fig. 3a). Reciprocally, the immunoprecipitation of endogenous 14-3-3 proteins using anti-14-3-3 antibodies confirmed selective coimmunoprecipitation of W and mW-CTD proteins (Fig. 3c). Importantly, the W protein was unable to IP exogenous hemagglutinin (HA)-tagged p65, suggesting that the W protein and p65 do not interact with each other directly (Fig. 3a, last lanes). Instead, the interaction between the W protein and 14-3-3 was somewhat reduced in the presence of the overexpressed NF-κB p65, possibly reflecting the competition between W and p65 proteins for binding 14-3-3.

We next analyzed the interaction between the W protein and 14-3-3 in an infectious context using either *wt* NiV, rNiV, or rNiV deficient for the CTD of the W protein (rNiV-W$_{\Delta CTD}$) (Fig. 3b). We examined whether W and 14-3-3 proteins interact during infection in two different cell types: HeLa cells used in previous experiments, and the human pulmonary microvascular endothelial cell (HPMEC) line chosen because endothelial cells are natural targets of NiV infection. In both wt NiV and rNiV-infected cells, the W protein was retrieved in the eluate of immunoprecipitated endogenous 14-3-3, confirming the interaction between the W protein and 14-3-3 in NiV-infected cells. As expected, no protein reacting with α-W-CTD could be detected after infection with rNiV-W$_{\Delta CTD}$.

Next, 14-3-3 proteins interact with their cellular partners via phosphorylated serine or threonine-containing motifs located within intrinsically disordered regions[44]. The 14-3-3-Pred web tool[45] predicted with a high score two putative binding sites, FTLRNL[**S199**]DPAK and VSMRRM[**S449**]N, located within the NTD and CTD of W, respectively. The Ala substitution of S449 (W-S449A) alone or in combination with an Ala substitution of S199 (W-S199A-S449A) resulted in the loss of coimmunoprecipitation of endogenous 14-3-3-proteins in both HeLa cells and HPMECs, whereas the W-S199A variant was as efficient as the *wt* W protein in precipitating 14-3-3 proteins. These results were confirmed by reverse coimmunoprecipitation with 14-3-3 proteins as baits (Fig. 3c). Overall, these data indicate that during NiV infection, the nonstructural W protein binds to endogenous 14-3-3 via the predicted binding site VSMRRM[**S449**]N located within the mW-CTD.

To investigate the importance of W protein phosphorylation for its interaction with 14-3-3, cellular extracts from HPMECs expressing FLAG-tagged wt or truncated/variant W proteins were treated with Lambda phosphatase to dephosphorylate the proteins prior to immunoprecipitation. This resulted in a loss of interaction between W and 14-3-3 proteins, demonstrating that the latter cannot take place without W protein phosphorylation (Supplementary Fig. 4) in agreement with the recent observation that the W-Ser499 phosphate group makes key bonds with the binding groove of 14-3-3 proteins in a co-crystal structure[42].

**W-S449 is necessary for blocking NF-κB activation**. We next addressed the functional consequence of 14-3-3 binding to the W protein on the NF-κB response. Similar to the W and mW-CTD

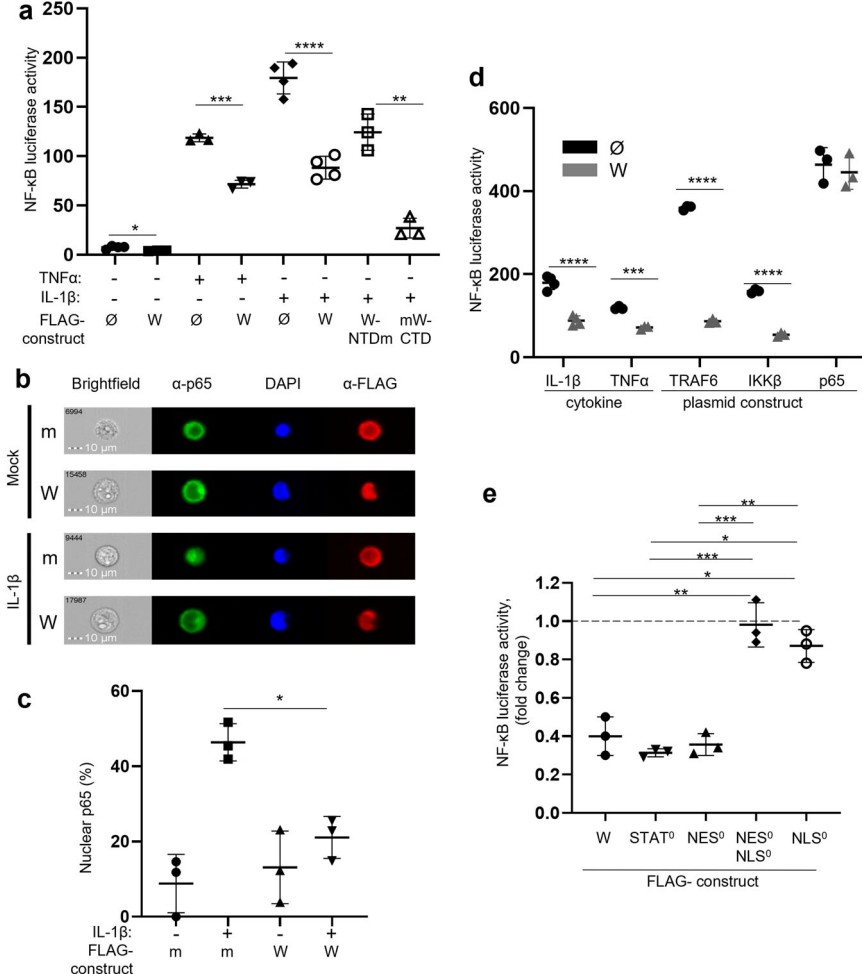

**Fig. 2 W inhibits the canonical NF-κB pathway downstream of IKKβ by a mechanism depending on its nuclear import and inhibits nuclear accumulation of p65. a** HeLa cells were transfected with a NF-κB_luc plasmid together with either an empty vector (Ø) or plasmids encoding FLAG-tagged W protein variants. Transfection with *Renilla* luciferase was used for normalizing the results. Cells were stimulated 20 h after transfection with 10 ng/mL of TNFα or IL-1β for 4 h prior to measurement of NF-κB activity by luminescence quantification. **b**, **c** HeLa cells were transfected with FLAG-mCherry (m) or FLAG-tagged W (W) protein encoding plasmid and stimulated or not with 10 ng/mL IL-1β for 20 min. Cells were immunostained for NF-κB p65, FLAG, and DAPI, and the percentage of cells displaying nuclear localization of p65 was determined using ImageStreamX. **d** Cells were transfected with an empty vector (Ø) or a plasmid encoding the FLAG-tagged W protein and stimulated with either IL-1β or TNFα as detailed for **a**, or stimulated by transfection with a plasmid encoding either TRAF6, IKKβ, or NF-κB p65. **e** Cells were transfected with the NF-κB_luc vector together with the indicated plasmids encoding FLAG-tagged W protein or variants of it harboring disabled STAT1/4-binding site, NES, and/or NLS indicated as STAT$^0$, NES$^0$, NES$^0$NLS$^0$, and NLS$^0$, respectively, and stimulated 20 h after transfection with 10 ng/mL of IL-1β. The data were expressed as the fold change of the signal observed with the empty vector (dotted line). NF-κB activity was assessed by luminescence quantification. The data represent mean values of three independent experiments showed as mean ± SD. Pairwise comparisons were carried out using a Student's *t*-test and multiple comparisons were performed using one-way ANOVA completed by a Tukey's multiple comparisons test (*$p < 0.05$, **$p < 0.01$ ***$p < 0.001$, ****$p < 0.0001$).

proteins, W-S199A significantly inhibited IL-1β-induced NF-κB activation in HeLa cells and HPMECs, whereas the expression of W-S449A, W-S199A-S449A, and W-NTDm proteins did not affect the IL-1β-induced NF-κB response (Fig. 4a and Supplementary Fig. 3b). This loss of inhibitory function correlated with a reduction in the accumulation of p65 phosphorylated at Ser536 (phospho-p65), which is known to be a hallmark of the NF-κB pathway activation[4] and to govern IL-8 transcription[46]. Indeed, IL-1β-induced stimulation of cells transfected with W, W-S199A, or mW-CTD expression vectors resulted in lower levels of nuclear phospho-p65 compared to those observed in cells transfected with vectors expressing W-S449A, W-S199A-S449A, or W-NTDm, or an empty vector (Ø) (Fig. 4b and Supplementary Fig. 5). Moreover, fluorescent intensity analysis across the diameter of transfected cells demonstrated that NF-κB p65 did not

accumulate in the nucleus of IL-1β-stimulated cells when W, W-S199A, or mW-CTD proteins were expressed. In contrast, the expression of W-S449A, W-S199A-S449A, or W-NTDm proteins did not affect the IL-1β-induced p65 intranuclear translocation (Fig. 4c, d).

In order to analyze the importance of 14-3-3 in the W protein-mediated inhibition of p65 nuclear translocation, we generated a plasmid coding for the R18 peptide, an inhibitor of 14-3-3[47]. Expression of HA-tagged-R18 in HPMECs disabled the inhibitory effect of W, mW-CTD, and W-S199A proteins on NF-κB nuclear translocation induced by IL-1β (Fig. 5). Based on these findings, we conclude that the interaction of the W protein with 14-3-3 inhibits the NF-κB response by preventing the accumulation of both p65 and its transcriptionally active form, p65-Ser536$^P$, in the nucleus.

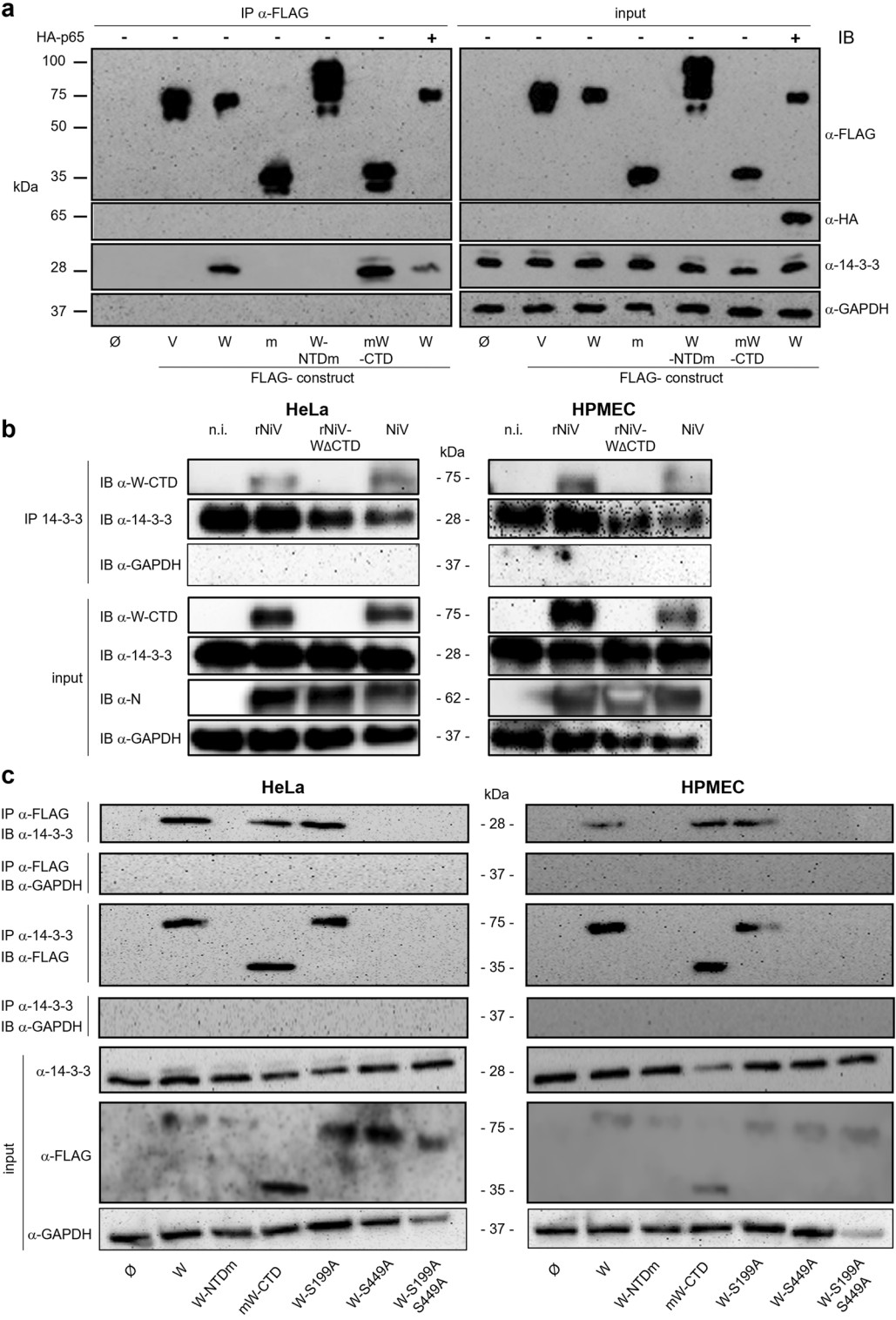

**Fig. 3 W protein requires CTD-S499 to interact with 14-3-3 in transfected and infected cells.** HeLa cells (**a–c**) or HPMECs (**b**, **c**) were transfected with a plasmid encoding FLAG-tagged V or truncated/variant W proteins, fused or not with mCherry (shortened as m), or HA-p65; an empty vector (Ø) and a plasmid encoding FLAG-mCherry (m) were used as controls. **a** Coimmunoprecipitation of endogenous 14-3-3 protein or exogenous HA-p65 with FLAG-tag NiV proteins (IP) using anti-FLAG antibodies bound to magnetic beads and detected by western blotting (IB) using rabbit anti-pan 14-3-3 or mouse anti-HA antibodies. **b** HeLa and HPMECs were mock-infected (n.i.) or infected with rNiV, rNiV-W$_{\Delta CTD}$, or *wt* NiV at MOI = 3. Cells were lysed 24 h later and immunoprecipitation was performed using anti-pan 14-3-3 antibodies coupled to magnetic beads. Input cell extracts and the eluate from beads were analyzed by western blotting using anti-NiV N, anti-W-CTD, anti-14-3-3, and anti-GAPDH antibodies. **c** Coimmunoprecipitation of endogenous 14-3-3 and FLAG-tagged W and W variant proteins in transfected HeLa and HPMECs was performed using either anti-pan 14-3-3 or anti-FLAG antibodies, and analyzed by western blotting as in **a**.

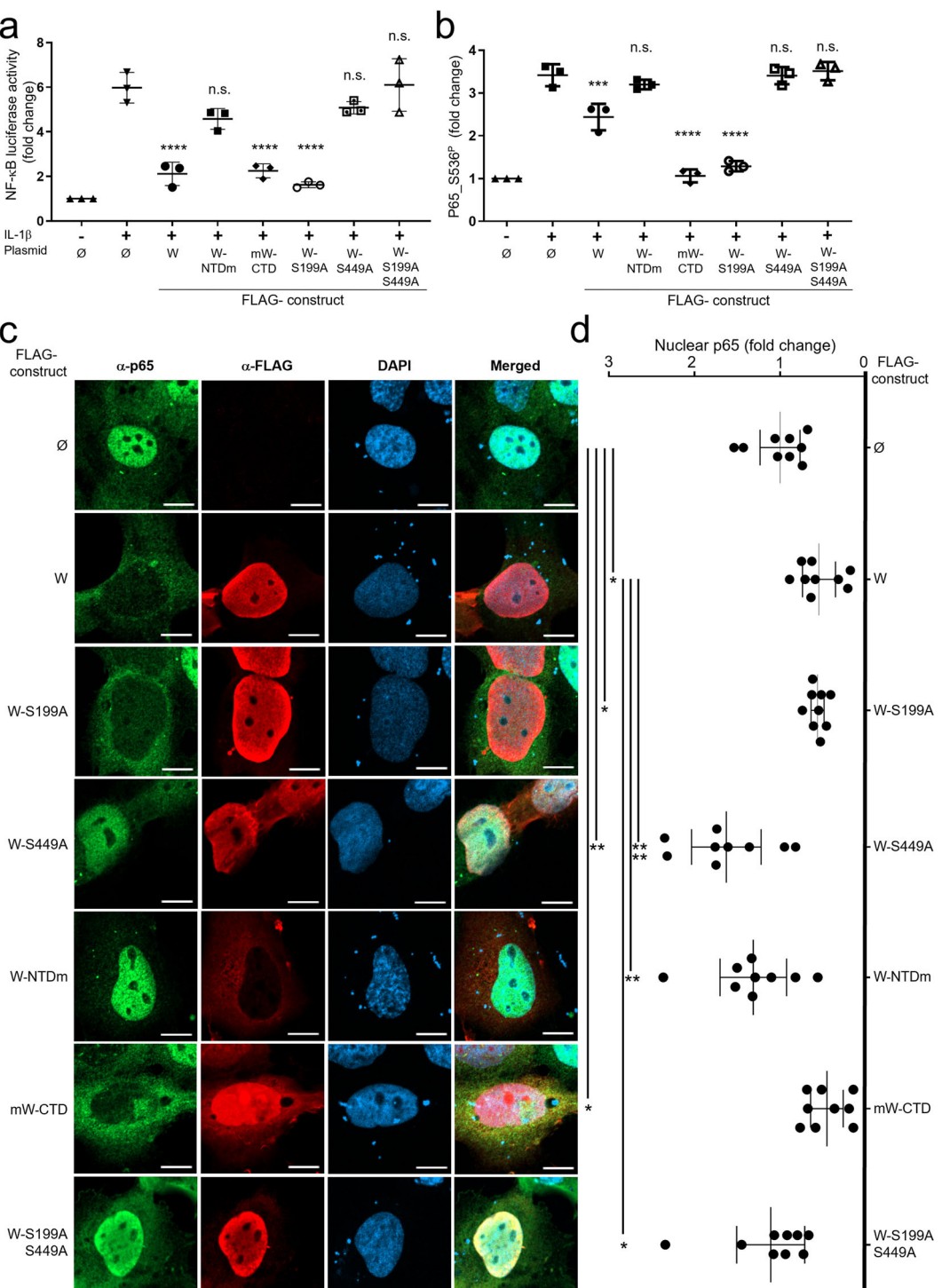

**Fig. 4 W-S449 is required for the inhibition of the canonical NF-κB pathway, and the phosphorylation and nuclear accumulation of NF-κB p65. a** HPMECs were cotransfected with a NF-κB_luc plasmid and a plasmid expressing the indicated FLAG-tagged W protein. Transfection with *Renilla* luciferase was used for the normalization of obtained results. Twenty hours after transfection, cells were stimulated with 10 ng/mL of IL-1β for 20 min and NF-κB activity was assessed by luminescence quantification. The data represent mean values of three independent experiments showed as mean ± SD". **b**, HPMECs were transfected with the indicated plasmids. Forty-eight hours after transfection, cells were stimulated with 10 ng/mL of IL-1β for 20 min, and p65 and phosphorylated p65 at Ser536 (p65_Ser536$^P$) were evaluated by western blotting. Data columns represent the mean ± SD of densitometry measurements done on samples from three independent experiments using ImageJ software. **c** HPMECs were transfected with the indicated plasmids and stimulated with 10 ng/mL of IL-1β for 20 min. After fixation and staining with NF-κB p65 and anti-FLAG antibodies, cells were analyzed by confocal microscopy (scale bar = 10 μm). **d** Mean fluorescence intensity of nuclear p65 protein. The nuclear signal obtained from anti-FLAG labeled cells was normalized to the nuclear signal from unlabeled cells and expressed as fold change. Error bars represent the confidence interval of the mean (CI 95%) for nine cells combined from three independent experiments. Statistical significance was assessed by one-way unpaired ANOVA completed with a multiple-comparisons Tukey's test; only significant comparisons are shown (*$p < 0.05$, **$p < 0.01$, ***$p < 0.001$, ****$p < 0.0001$).

**The W protein induces nuclear accumulation of 14-3-3, both in vitro and in vivo, and inhibits the production of proinflammatory cytokines**. We then studied the effect of the W protein on the intracellular distribution of 14-3-3. In the absence of NF-κB activation by a cytokine, the endogenous 14-3-3 accumulated into the nucleus upon the expression of W protein, whereas 14-3-3 was equally distributed between the cytoplasm and the nucleus of cells either left untreated (n.t.) or transfected with an empty plasmid (Ø) (Supplementary Fig. 6). The ability of W protein to re-localize 14-3-3 in the nucleus relies on W-CTD-S499 with the ability of each W protein variant recapitulating the effect seen on p65 upon stimulation of the cells by IL-1β (Figs. 4 and 5). Importantly, the W-S449A or W-S199A-S449A proteins that accumulated into the nucleus were unable to re-localize 14-3-3 in the cell nucleus in agreement with their inability to interact with 14-3-3 proteins (Supplementary Fig. 6). Similar observations were made upon NF-κB activation by IL-1β (Fig. 6a, b and Supplementary Fig. 7). These data demonstrate that the W protein induces nuclear accumulation of 14-3-3 regardless of the cell's activation level. Correlatively, nuclear accumulation of 14-3-3 was observed only when the W protein had an intact NLS. Then, 14-3-3 retained a diffuse distribution between the cytoplasm and the nucleus in cells expressing the W-NLS$^0$ or W-NES$^0$NLS$^0$ but not the W-NES$^0$ protein variant (Supplementary Fig. 8).

We analyzed whether the accumulation of 14-3-3 in the nucleus could also be observed in vivo in a context of infection. NiV-infected AGMs rapidly developed interstitial pneumonia. The presence of NiV-infected cells in lungs and kidneys was confirmed by necropsy[19]. The examination of lung sections by confocal microscopy and the quantification of the density profile of cells immunolabeled with anti-W-CTD antibodies revealed nuclear accumulation of both W and 14-3-3 proteins (Fig. 6c, d). In contrast, 14-3-3 was diffusely distributed in both the nucleus and the cytoplasm of surrounding cells not labeled with anti-W-CTD (i.e., cells that were not infected) as observed in lung cells from non-infected animals (Fig. 6c, d). Therefore, the W protein induces nuclear accumulation of 14-3-3, both by acting through its 14-3-3-binding site and NLS.

Finally, we analyzed the role of the W protein in modulating proinflammatory cytokine expression in human endothelial cells. Following IL-1β stimulation, the IL-6 and IL-8 transcriptional responses were inhibited in HPMECs transfected to express W, mW-CTD, or W-S199A proteins. In contrast, expression of variants unable to bind to 14-3-3, i.e., W-NTDm and W-S449A proteins, did not alter cytokine transcription levels (Fig. 6e, f). This pattern is in complete agreement with data obtained using the NF-κB reporter gene assay (Fig. 4a) and reflects the 14-3-3 nuclear accumulation observed in endothelial cells (Fig. 6a, b). It also falls in line with previously published data demonstrating that the mW-CTD is involved in the inhibition of IL-6 and IL-8 production during NiV infection in HPMECs[24]. Altogether, these results indicate that the W protein participates in the down-modulation of the NF-κB pathway in infected cells and thus could contribute to the dysregulation of the inflammatory response during NiV infection.

## Discussion

This study reveals the ability of the nonstructural W protein of NiV to down-modulate NF-κB signaling and the subsequent activation of proinflammatory cytokines via 14-3-3-mediated exclusion of the NF-κB p65 subunit from the nucleus. Our data corroborating the capacity of the W protein to hijack the NF-κB pathway are summarized in Supplementary Table 2. In concert with our results, a study published during the writing of this

manuscript demonstrated the ability of NiV W protein to bind to all isoforms of 14-3-3 via the phosphorylated S449 by performing binding assays and resolving the three-dimensional (3D) structure of co-crystallized 14-3-3 in complex with a C-terminal peptide of W protein[42]. Importantly, the maintenance of nuclear localization of the W-S449A variant protein (Figs. 4c and 6a) could be explained by the preservation of its binding to importin α3 described in that report. Indeed, Henipavirus W proteins[28,42,48] and NF-κB p65/p50 complexes[49] are imported into the nucleus by binding to importin α molecules. The fact that W and p65/p50 proteins preferentially use different subsets of importins (importin α3, α4, and, with a lower affinity, α1 for W, vs. α3, α4, α5, and α6 for p65/p50) further indicates that the W protein does not act simply by competing with p65/p50 for binding importins. Furthermore, as W proteins from both NiV and Hendra virus (HeV), another highly pathogenic emergent *Henipavirus*, were shown to bind 14-3-3[42], the inhibition of NF-κB signaling by NiV W protein demonstrated here may also apply to HeV W protein, as the 14-3-3-binding site in W-CTD is identical between the two Henipavirus (see Supplementary Fig. 1 for amino acid sequence alignments).

14-3-3 is a family of ≈30 kDa α-helical homo- or heterodimeric proteins highly conserved in eukaryotes from yeast to mammals[43]. They act as scaffold and adaptor proteins, and regulate numerous signaling pathways via direct interactions with partners harboring a phosphorylated 14-3-3-binding motif[50]. In particular, 14-3-3 is involved in cyto-nuclear trafficking of many cellular factors involved in regulating transcription and chromatin remodeling. They can act at least at three different levels. First, 14-3-3 can prevent the nuclear import of a factor by shielding its NLS, as in the case of Cdc25[51,52]. Second, 14-3-3 can inhibit the DNA-binding ability of a factor right after its phosphorylation in the nucleus, as documented for the human Forkhead transcription factor (FKHRL) and its *Caenorhabditis elegans* homolog DAF-16, and thus induce their export from the nucleus. Finally, 14-3-3 can force the nuclear export of a ligand by enabling the function of its NES by an unknown mechanism, as reported for FOXO1[53], class II histone deacetylases[54], TAZ[55], COP1[56], telomerase[57], and the IκBα/p65 complex[10]. Normally, 14-3-3 intensively shuttles in and out of the nucleus[58]. Following TNFα stimulation, it rapidly enters the nucleus and interacts with p65 and IκBα, leading to the dissociation of p65 from chromatin and its return to the cytoplasm after 60 min. Thus, 14-3-3 participates in negative feedback regulation of NF-κB signaling. Interestingly, 14-3-3 can interact with p65 and IκBα simultaneously, as indicated by its ability to bind p65 and IκBα alone and/or in complex in vitro[10,59], and the recently solved 3D structure of 14-3-3 bound to IκBα-phosphorylated binding motif[60]. Furthermore, IκBα, which can freely enter the nucleus, acts as a repressor of NF-κB-dependent transcription by promoting dissociation of p65/p50 from the DNA and its export from the nucleus as a protein complex[61].

By combining our results with data available in literature, we propose a 14-3-3-dependent mechanistic model of W-mediated inhibition of NF-κB-induced proinflammatory response (Fig. 7). According to this model, the W protein accelerates the basal turnover of the NF-κB transcription factor (Fig. 7a) by increasing the intranuclear accumulation of 14-3-3 (Fig. 7b). Due to the higher affinity of importins to the W-NLS, which overlaps with its 14-3-3-binding site[42], the W protein is imported into the cell nucleus and accumulates there. Then, its phosphorylated form binds to 14-3-3 molecules entering the nucleus, thus skewing their nucleo-cytoplasmic distribution in favor of the nucleus (Fig. 7). This increases the probability of 14-3-3 recruitment to the IκBα/p65 complex, possibly due to its higher avidity for the latter than for the W protein. In favor of this hypothesis, a lower

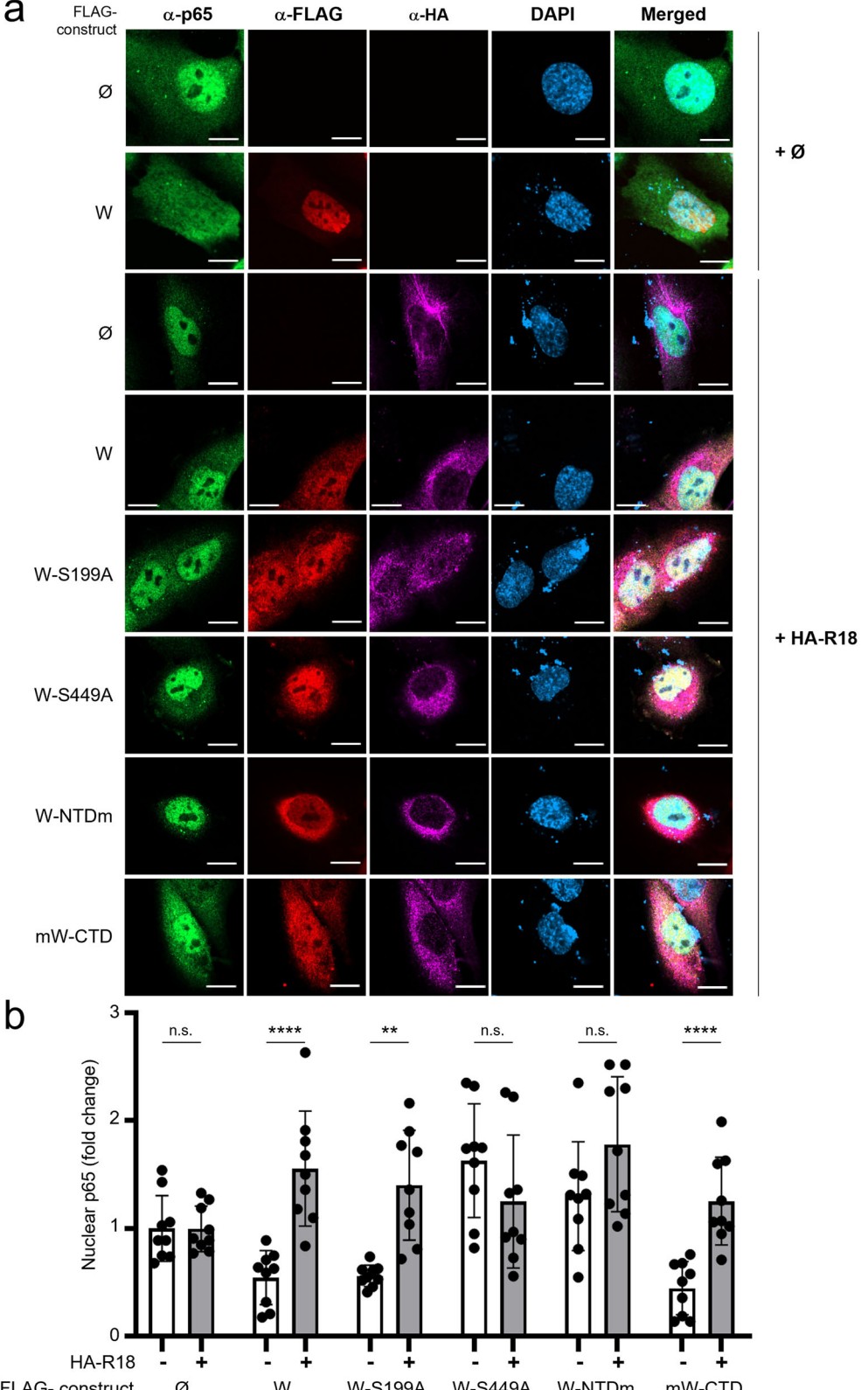

**Fig. 5 Expression of the 14-3-3 inhibitory peptide R18 disables W protein induced inhibition of NF-κB p65 nuclear localization. a** Representative images of HPMEC cells transfected or not (Ø) with plasmids encoding HA-R18 and indicated FLAG-tagged W proteins, and stimulated with 10 ng/mL of IL-1β for 20 min (scale bar = 10 μm). Cells were fixed and stained for NF-κB p65, anti-HA, and anti-FLAG, and analyzed by confocal microscopy. **b** The mean fluorescence intensity for p65 protein was measured in the cell nucleus. Signal obtained from W expressing cells (as labeled by anti-FLAG) was normalized to the signal from cells not labeled by anti-FLAG and expressed in fold change. Error bars represent the mean's confidence interval (CI 95%) for nine cells, combined from three independent experiments. Each combination has been done in three different wells performed in two independent experiments. Statistical significance was assessed by a one-way unpaired ANOVA, multiple comparisons Tukey's test; **\*\*p < 0.01 and \*\*\*\*p < 0.0001.

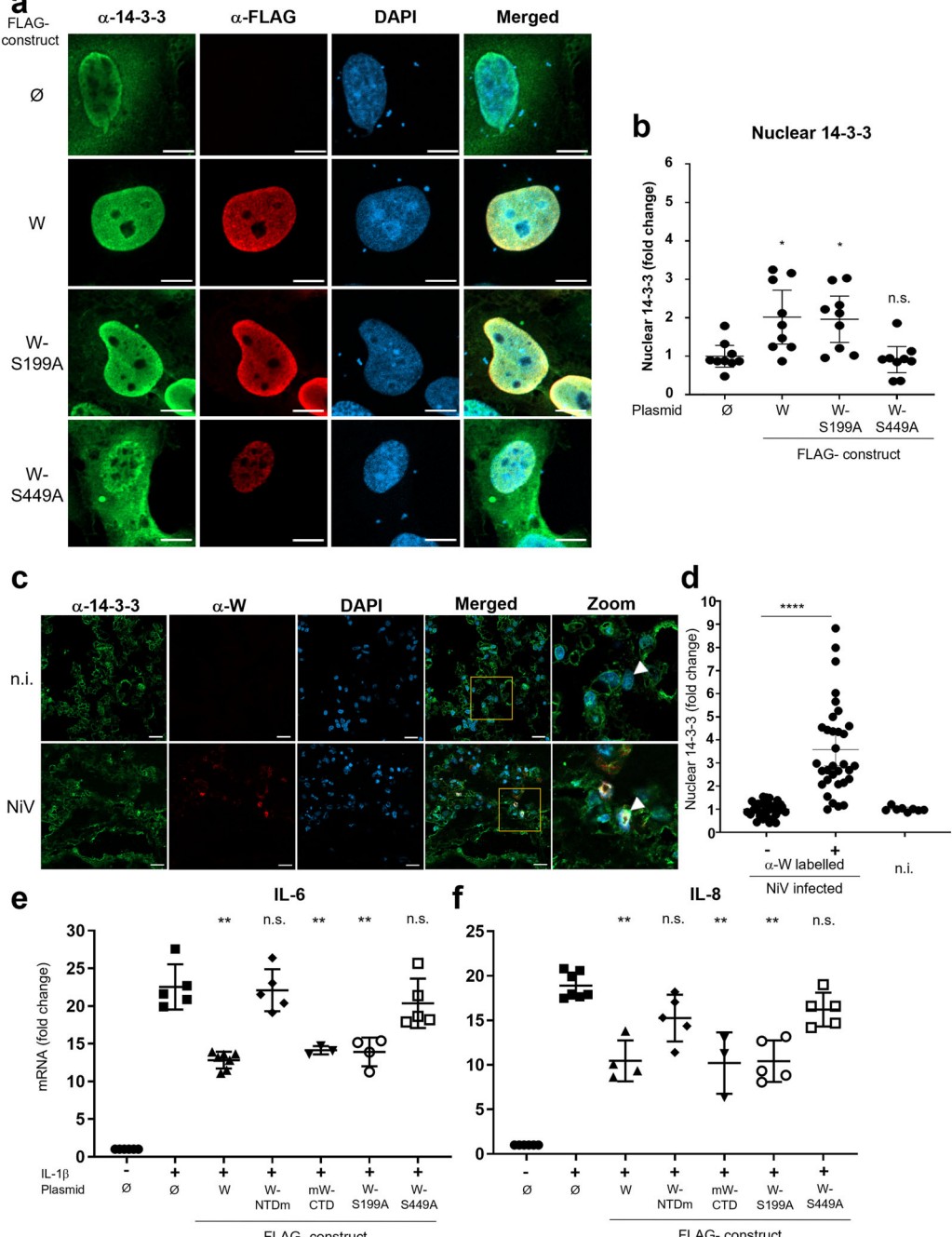

**Fig. 6 W protein induces 14-3-3 nuclear accumulation in transfected and in NiV-infected cells, and suppresses the production of proinflammatory cytokines in vitro. a** HPMECs were transfected with plasmids encoding FLAG-tagged W protein variants. Twenty hours later, cells were stimulated with 10 ng/mL of IL-1β for 20 min, fixed, permeabilized, stained with DAPI and anti-FLAG and anti-14-3-3 antibodies before analysis by confocal microscopy (scale bar = 10 μm). **b** Mean fluorescence intensity for 14-3-3 protein was measured in a IL-1β stimulated cell nucleus. Signal obtained from anti-W-labeled cells was normalized to the signal from unlabeled cells and expressed in terms of fold change. Error bars represent the confidence interval of the mean (CI 95%) for nine cells, combined from two independent experiments. Statistical significance was assessed by a one-way unpaired ANOVA completed with a multiple-comparisons Tukey's test; *$p < 0.05$. **c** Nuclear accumulation of 14-3-3 in NiV antigen-positive cells from lungs of in vivo-infected AGM with a representative image including an enlarged portion of the merged images. A NiV-infected AGM was killed 10 days post infection and lung sections were labeled using DAPI, anti-W-CTD (red), and pan 14-3-3-specific rabbit polyclonal antibodies (scale bars = 20 μm). **d** Mean nuclear signal of 14-3-3 in infected cells as labeled with anti-W-CTD antibodies, expressed as the fold change of nuclear signal in non-infected cells as lacking W-CTD labeling within the same field. Error bars represent the confidence interval of the mean (CI 95%) from five different lung sections. Statistical significance was assessed by paired $t$-test between unlabeled and anti-W-CTD-labeled cells from NiV-infected slices (****$p < 0.0001$). Mean nuclear 14-3-3 signal in the nucleus of lung cells from a non-infected AGM (n.i.) was comparable to that observed in non-infected cells (i.e., lacking anti-W-CTD labeling) from NiV-infected AGM. **e**, **f** HPMECs were transfected with plasmids encoding FLAG-tagged W protein variants and stimulated 20 h later with 10 ng/mL of IL-1β for 4 h. Cells were lysed 24 h later and quantified for their contents in IL-6 and IL-8 mRNA by RT-qPCR. Data columns represent mean values ± SD of at least three independent experiments with each point done in triplicate (**$p < 0.01$ using ordinary one-way ANOVA followed by a Turkey's multiple comparisons test).

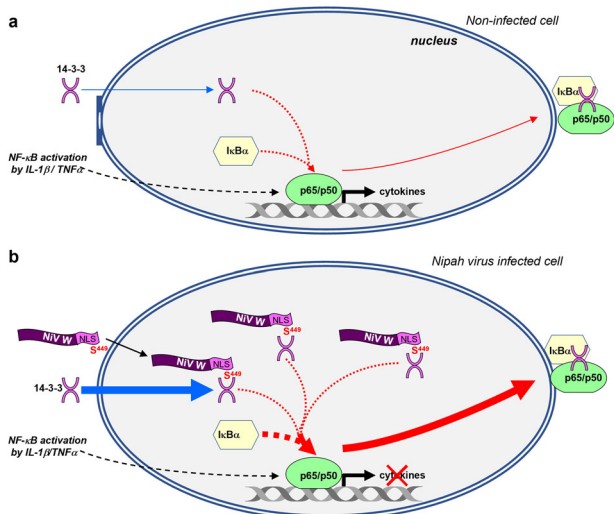

**Fig. 7 Model of the W-mediated inhibition of the NF-κB pathway. a** IL-1β/TNFα stimulation (black arrows) leads to the entry of NF-κB p65/p50 complex into the cell nucleus and the activation of cytokine expression. The activation is controlled by regulatory proteins 14-3-3 and IκB-α (red arrows), resulting in the nuclear export of 14-3-3/p65/p50-IκBα complex[7,10,60]. 14-3-3 remains in equilibrium between the cytoplasm and the nucleus[58]. **b** NiV W protein is imported into the nucleus of the infected cell, mediated by its NLS within its CTD. Via its CTD-S449, the W protein binds to 14-3-3, resulting in an increased influx of 14-3-3 into the nucleus (thicker blue arrow). When the W protein is artificially debilitated by either NLS mutation or S449A substitution, it loses, respectively, its ability to enter into the nucleus, or to interact with nuclear 14-3-3 and, consequently, to inhibit NF-κB activity (not illustrated). Accumulated nuclear 14-3-3 enhances the export of the p65/p50/IκB-α complex from the nucleus (thicker red arrow). The increased translocation rate of NF-κB p65/p50 out of the nucleus prevents its efficient binding onto the promoters of proinflammatory cytokines and results in the inhibition of the host inflammatory response. Only details of the intranuclear part of the NF-κB pathway, which are critical for the comprehension of the action of the W protein revealed in this study, are presented in the figure. Of note, as the molecular mechanism used by 14-3-3 molecules to enter the nucleus (passively or actively) is still unknown, the 14-3-3 nuclear entry trafficking is not indicated by the classical double inverted arrows used to mean (dis)equilibrium. Additional details are described in the Discussion.

amount of endogenous 14-3-3 was co-immunoprecipitated with the W protein in cells overexpressing p65 (Fig. 3a). Then, this could explain the inability of the W protein to inhibit NF-κB activation by overexpressed p65 (Fig. 2d). In turn, the probability of NF-κB dissociation from the DNA increases, leading to a repression of cytokine gene transactivation and an efflux of IκBα/p65/p50 from the nucleus[10,60]. According to this model, the W protein could reinforce the physiological repressing effect of 14-3-3 on NF-κB transcriptional activity by forcing nuclear accumulation of 14-3-3 in agreement with 14-3-3, ensuring the efficient nuclear export of NF-κB p65/p50 complexes[10] and the pleiotropic inhibitory effect the W protein on the cellular gene transcription recently reported[42]. Globally, our proposed mechanism fits with the current knowledge regarding the interaction of 14-3-3 with p65 regulated by phosphorylation of the later. Although the W protein does not interact with p65 and seems rather to compete out with p65 for binding to 14.3.3 within the nucleus, we cannot exclude that W protein can alter the phosphorylation profile of p65. Indeed, the nuclear trafficking of p65 is finely tuned by successive phosphorylation steps: phosphorylation on the residue S536 of p65 enables its nuclear localization (and transactivation

activity)[5], then it is exported outside the nucleus p65 upon its phosphorylation at positions S42, S281, and S340, which allows its association with 14-3-3 and IκBα as cargo export machinery[10]. Our model raises challenging questions that merit future investigation: how and where is the W protein phosphorylated at S449? What are the respective affinities/avidities of 14-3-3 for the (phosphorylated) W and IκBα/p65(/p50) proteins? Could the W protein interfere with kinases and/or phosphatases controlling the phosphorylation status of p65? As 14-3-3 mediates the down-regulation of numerous genes, W-mediated accumulation of 14-3-3 proteins in the cell nucleus is expected to unbalance a range of transcriptional pathways. This may explain the pleiotropic dysregulation of genes belonging to several Gene Ontology families recently reported with regards to Henipavirus W-S449[42].

The W proteins of NiV and HeV are particular among the *Paramyxoviridae* family due to their unique CTD sequence. Due to the absence of an editing site, neither V nor W proteins could be produced following infection with Cedar virus, an apparently nonpathogenic Henipavirus[62]. Therefore, this additional anti-inflammatory feature of the W protein may be linked to the extreme pathogenicity of NiV and HeV. Indeed, NiV interferes with both the adaptive and the innate immune response, restraining the efficiency of antiviral defense during infection (reviewed in ref. [19]). The intrinsically disordered nature of W[63] confers it certain structural flexibility, favoring multiple interactions with different binding partners, including STATs. However, NiV-mediated antagonism of STAT1 and inhibition of IFN-I signaling, which it shares with the members of the *Morbillivirus* genus[64], seem to play a minor role in NiV-mediated disease in the ferret model[65].

14-3-3 scaffold proteins are known to regulate numerous signaling pathways including NF-κB. As NF-κB is a critical regulator of innate immunity and inflammation[7], the interaction of the W protein with 14-3-3 proteins may play an important role in Henipavirus invasiveness and pathogenicity. The harnessing of 14-3-3 by a viral protein represents an additional molecular mechanism of dampening the NF-κB pathway deployed by a pathogenic virus. As both the NF-κB pathway and the 14-3-3 family are targets for drug development[43], these results may pave the way for developing novel antiviral approaches. Future studies of W protein functions should focus on questions still pending, including the mechanism by which it inhibits innate immunity, particularly in the context of its natural reservoir, fruit bats, which appear tolerant to NiV infection. Indeed, the particularities observed in bat NF-κB pathways[66,67] suggest that the W protein may act differently in bats.

## Methods

**Cell lines.** Cells of the human carcinoma HeLa cell line (Cat# ATCC CCL-2) were grown in Dulbecco's modified Eagle's medium (DMEM) + GlutaMAX (Life Technologies™, Cat# 61965-026) supplemented with 10% of heat-inactivated (30 min at 56 °C) fetal bovine serum (fetal calf serum, FCS) (Dutscher, Cat# S1810-500), and 15 μg/mL Gentamicin (Life Technologies™, Cat# 15750-037). HEK293T cells stably transduced with a NF-κB_luc reporter gene[68] were grown in the same medium supplemented with 200 μg/mL Hygromycin (Invivogen™, Cat# ant-hg-1). HPMEC cells[69] were cultured in Endothelial Cell Growth Medium (PromoCell®, Cat# C-22121) in flasks coated with 0.1% bovine gelatin in phosphate-buffered saline (PBS) (Gibco, Cat# 14190-094). All cell types were incubated at 37 °C with 5% $CO_2$ and were tested negative for *Mycoplasma* spp.

**Viruses.** Wild-type NiV Bangladesh isolate SPB200401066 (CDC, Atlanta, USA; GenBank AY988601) (*wt* NiV), the rNiV, and rNiV in which an in-frame stop codon was added to limit translation of the +2 frame to the common P/V/W-NTD and prevent the expression of the full-length NiV W protein (rNiV-W$_{\Delta CTD}$)[36,70] were prepared in Vero-E6 cells negative for *Mycoplasma* spp. as previously described[25]. NiV infections were carried out at the INSERM Jean Mérieux BSL4 laboratory in Lyon, France.

**Animal infection**. Three healthy 3-year-old AGMs from Saint Kitts were infected intratracheally in the BSL4 laboratory (Jean Mérieux, Lyon) with $10^2$ Plaque forming units of *wt* NiV in 2 mL of DMEM under anesthesia with Zolétil®. Animals were followed daily and kept under video-camera observation. Blood sampling was performed on days: 0, 2, 4, 6, and 8, using EDTA-containing tubes. Samples were centrifuged at $1500 \times g$ for 10 min and plasma samples were kept at −80 °C. Killing was performed when the critical point of infection was reached, 8–10 days after infection, by injecting of 5 mL of intracardiac Doléthal®. Necropsies were performed and organs were fixed using 4% formaldehyde (Thermo Scientific, Cat# 28908) during 14 days in BSL4 conditions and then processed for histopathological analysis.

**Ethical statement**. Animals were handled in strict accordance with good animal practice as defined by the French national charter on the ethics of animal experimentation and all efforts were made to minimize suffering. Animal work was approved by the Regional ethical committee and French Ministry of High Education and Research, and experiments were performed in the INSERM Jean Mérieux BSL4 laboratory in Lyon, France (French Animal Regulation Committee Number D69 387 05 02).

**Cytokine/chemokine multiplex assay**. Circulation levels of cytokines/chemokines in NiV B-infected AGMs were measured in duplicate plasma samples (25 µL) in the BSL4 laboratory INSERM Jean Mérieux, using Non-Human Primate Cytokine Magnetic Bead Panel (Merck, Cat# PCYTMG-40K-PX23 and Cat# PRCYT2MAG40K). Plates were prepared according to the manufacturer's recommendations and samples were assayed for median intensity fluorescence across an at least 50-bead region using Magpix luminex (Merck, Millipore).

**Immunofluorescence analysis of formaldehyde-fixed tissue**. Lungs from AGMs were embedded in paraffin wax (Sigma Aldrich, Cat# P3558) and sectioned at 5 µm. Slides were deparaffinated and rehydrated in 3 Xylene (VWR, Cat# 28975.291) baths for 5 min each, followed by two 100% alcohol (VWR, 20821.296) baths for 5 min followed by multiple baths using a decreasing concentration of alcohol for 3 min each. After deparaffinization, slides were put into a handmade sodium citrate solution with 10 mM sodium citrate (Sigma™, Cat# C8532), 0.05% Tween 20 (VWR, Cat# 28829.296) in distilled water, in a boiling water bath for 20 min for heat-induced epitope retrieval, and washed in PBS three times for 3 min afterwards. Samples were then permeabilized and blocked for 20 min using a permeabilization and blocking solution (PBS, 3% bovine serum albumin (BSA), 0.15% Triton X-100). After incubation, slides were immunostained overnight at 4 °C with rabbit anti-14-3-3 pan antiserum (Merck Millipore™, Cat# AB9748-I) (1 : 100 in permeabilization and blocking solution). After three washes in PBS for 5 min, slides were incubated for 1 h at room temperature (RT) with Alexa 488-conjugated anti-rabbit antibody (Invitrogen™, Cat# A11034) (1 : 750 in PBS). Nuclei were stained with 4′,6-diamidino-2-phenylindole (DAPI) (Thermo Fisher™, Cat# 62248) (1 : 1000 in the mix of secondary antibodies). Slides were then washed five times in PBS 3% BSA (Sigma Aldrich, Cat# A3059) for 5 min, and then incubated overnight at 4 °C with biotinylated rabbit anti-NiV W-CTD (GeneScript™), diluted 1 : 100 in PBS. After five washes in PBS 3% BSA for 5 min each, slides were incubated for 45–60 min at RT with Alexa 647-conjugated streptavidin (Invitrogen™, Cat# S21374) (1 : 500 in PBS), washed in PBS five times for 5 min, and mounted with Fluoromount-G (SouthernBiotech™, Cat# 0100-01) before observation by confocal microscopy (Zeiss™ LSM800).

**Plasmids**. The firefly luciferase reporter gene *pGL3-NFκBx5_luc* (containing five repeats of the enhancer element sequence 5′-TGGGGACTTTCC-3′) has already been described[71]. Plasmid pRL-Null (Promega™, Cat# PR-E2271) was used as a *Renilla* luciferase expressing plasmid to normalize transfection efficiency. All expression plasmids were made by cloning sequences obtained by PCR in a pCG plasmid backbone using InFusion-mediated recombination (Takara, Cat# 638933). The parental pCG-FLAG and pCG-HA plasmids contained a FLAG or HA tag linked to a GlyGlyGlySerGly linker (sequence: 5′-GGTGGAGGATCCGGA-3′) downstream of a start codon associated with a Kozak sequence (5′-GCCACCATG-3′). mCherry (m) and NiV cDNAs (V and W: AF376747.1) were cloned in the pCG-FLAG vector, and TRAF6 (NM145803.2), IKKβ (NM001556.2), and NF-κB p65 (NM021975.3) cDNAs were cloned in the pCG-HA vector. The cDNA encoding W-NTD (aa 1–407) and W-CTD (aa 408–450) were subcloned in-frame downstream to the FLAG-mCherry tag and between FLAG and mCherry in the pCG plasmid to produce W-NTDm and mW-CTD proteins, respectively. Alanine substitutions were introduced in key positions of the FLAG-tagged W cDNA to obtain W-NES (L174A, L186A)[026], W-NLS (K439A, K440A, R442A)[028], W-NES[0]NLS[0] (L174A, L186A, K439A, K440A, R442A), W-STAT[0] (G121E), a STAT non-binding variant[27] W-S199A, W-S449A, and W-S199A-S449A. A pCG empty vector (Ø) (Addgene, Cat# 51476) was used as a control and to normalize the transfected DNA in cells. The synthetic cDNA (5′-ggtaCCC-CACTGTGTGTCCCCCGAGATCTTTCGTGGTTAGATTTAGAAGCAAA-TATGTGTTTACCCTAGtctaga-3′) encoding the 14-3-3 antagonist R18 peptide (PHCVPRDLSWLDLEANMCLP)[47] was amplified by PCR and subcloned into the

pCG-HA plasmid using the NEB HIFI DNA assembly kit (New England Biolab), following the manufacturer's protocol and recommendations.

**Luciferase reporter assays**. Ninety-six-well plates were coated with 50 µg/mL of poly-D-lysine (Sigma Aldrich, Cat# P7280) for 1 h at 37 °C, then washed with water (Versol™) and dried before seeding $2 \times 10^4$ HEK293T NF-κB_luc cells per well. Twenty hours later, cells were infected with rNiV or rNiV-W$_{\Delta CTD}$ at a multiplicity of infection (MOI) of 3 prepared in DMEM 0% FCS (Life Technologies™, Cat# 61965-026). The medium was replaced 3 h post infection with DMEM 10% FCS supplemented with 200 µg/mL Hygromycin (Invivogen™, Cat# ant-hg-1) and 1 µM of a fusion-inhibitor peptide (peptide VIKI-dPEG4-chol, preventing virus-induced cell fusion[72]), and cells were cultured for an additional 15 h. Then, cells were treated with 10 ng/mL of human TNFα (PeproTech™, Cat# 300-01A) for 4 h. Cells were lysed using the lysis buffer from the Luciferase Assay System and luciferase assays were performed according to the manufacturer's protocol (Promega™, Cat# E1501). The luminescence from each well was measured via 15 s per well acquisition using a microplate reader (Infinite 200 Pro, Lifesciences, Tecan™). Then, for each 100 µL lysate sample, 20 µL of 6× loading buffer (0.35 M Tris HCl pH 6.8, 10.3% SDS, 36% glycerol, 5% β-ME, 0.012% bromophenol blue) were added and the lysates were heated for 15 min at 96 °C. The lysates were also analyzed by western blotting for expression of the viral P, V, and W proteins (Rabbit anti-NiV P/V/W-NTD (polyclonal raised against amino acids 1–407 of P/V/W[37]) and glyceraldehyde 3-phosphate dehydrogenase (GAPDH) proteins (Mouse anti-GAPDH, Chemicon™, Cat# MAB374). For each condition, experiments were performed in triplicate and repeated in three separate experiments.

Luciferase Reporter Assays in the context of transfection were performed using HeLa ($1.5 \times 10^4$), HPMEC ($2 \times 10^4$), or HEK293T ($2 \times 10^4$) cells seeded in 96-well plates. Twenty hours later, cells were transfected with 50 ng of the Firefly luciferase reporter gene *pGL3-NFκBx5_luc*[71], 25 ng of *Renilla* luciferase pRL-Null for normalization, and a total of 200 ng of plasmids using JetPrime (Polyplus™, Cat# 114-15) according to the manufacturer's protocol. The medium was replaced 4 h post transfection and cells were grown for additional 16 h. Twenty hours post transfection, cells were treated with either 10 ng/mL of human IL-1β (PeproTech™, Cat# 200-01B) or human TNFα (PeproTech™, Cat# 300-01 A) for 4 h. Luciferase assays were performed using the Dual-Glo Luciferase Assay system according to the manufacturer's protocol (Promega™, Cat# E2920). The luminescence of each luciferase was measured by a 10 s acquisition per well using a microplate reader (Mithras LB940, Berthold™). For each well, NF-κB-dependent luciferase activity was obtained as the ratio of the Firefly luciferase activity on the *Renilla* luciferase activity. For each condition, experiments were performed in triplicate and repeated at least in three separate experiments.

**Co-immunoprecipitations**. HeLa or HPMECs ($5 \times 10^5$) were seeded in six-well plates. Twenty hours after seeding, cells were transfected with a total of 2.5 µg of various plasmids using JetPrime (Polyplus™, Cat# 114-15) or Lipofectamine 2000 (Thermo Fisher Scientific, Cat# 11668019) according to the manufacturer's protocol. Four hours post transfection, the medium was replaced and cells were grown for an additional 16 h. Alternatively, cells were infected with the appropriate dilution of *wt* NiV, rNiV, or rNiV-WΔCTD (MOI = 3) in DMEM 0% FCS (Life Technologies™, Cat# 61965-026). Twenty hours post transfection or post infection, cells were lysed in RIPA buffer (Pierce, Cat# 89901), supplemented with a cocktail of protease inhibitors (Thermo Scientific, Cat# 78444) for 30 min on ice and centrifuged for 10 min at 4 °C at $15,000 \times g$. To analyze the role of phosphorylation in the interaction between 14-3-3 proteins and NiV W, each sample was resuspended in RIPA buffer and supplemented with the lambda phosphatase buffer, MnCl$_2$, and 1000 units of Lambda phosphatase. Then, samples were incubated for 30 min at 30 °C before performing the co-immunoprecipitations. Supernatants were incubated with rabbit anti-14-3-3 pan anti-14-3-3 (Merck millipore™ Cat# AB9748-I) or rabbit anti-W antibodies (Rabbit anti-NiV W-CTD (amino acids 408–450, GeneScript™) for 2 h at 4 °C. Protein A/G agarose beads (Pierce, Cat# 20421) were added to the mix overnight at 4 °C. Beads were then washed three times in washing buffer (RIPA buffer supplemented with a cocktail of protease inhibitors) and proteins were eluted in 100 µL of elution buffer (reducing agent 10× (Invitrogen, Cat# NP0009), Laemmli 4× (Life Technologies, Cat# NP0008), RIPA buffer, supplemented with a cocktail of protease inhibitors) for 15 min at 96 °C. Then the eluate and input cell extract samples were analyzed by polyacrylamide gel electrophoresis (SDS-PAGE) and western blotting.

For MS analysis, cells were lysed in 200 µL of lysis buffer (50 mM Tris HCl pH 7.4, 150 mM NaCl, 1 mM EDTA, 0.5% NP40, 50 mM NaF, 2 mM Na$_3$VO$_4$, 5% glycerol) with complete protease inhibitors (Roche™, Cat# 11836145001), incubated for 20 min on ice, and centrifuged for 15 min at 4 °C at $15,000 \times g$. The supernatant was incubated with anti-FLAG antibody-coated magnetic beads (Sigma™, Cat# M8823) for 2 h. The beads were then washed five times in washing buffer (50 mM Tris HCl pH 7.4, 150 mM NaCl, 1 mM EDTA, 0.05% NP40, 50 mM NaF, 2 mM Na$_3$VO$_4$, 5% glycerol) with complete protease inhibitors, and FLAG-tagged proteins were eluted for 1 h with 50 µL of washing buffer containing FLAG peptide 3× (Sigma™, Cat# F4799) at 240 ng/µL. To 50 µL of each protein sample, 10 µL of 6× loading buffer (0.35 M Tris HCl pH 6.8, 10.3% SDS, 36% glycerol, 5% β-ME, 0.012% bromophenol blue) were added and the samples were heated for 3 min at 100 °C. They were then analyzed by MS.

**Mass spectrometry**. Bands of stacking gels were lysed by trypsin and analyzed by liquid chromatography-MS/MS. Peptide separation was achieved on a C18 Pre-column (Acclaim$^{TM}$ PepMap$^{TM}$ 5 μm; 100 Å; 300 μm × 5 mm) and a C18 nano-Column (Accucore$^{TM}$ 2.6 μm; 150 Å; 75 μm × 500 mm) using a gradient from 5% acetonitrile, 0.1% formic acid to 50% acetonitrile, 0.1% formic acid in 120 min at 200 nL/min flow rate on a nanoLC Ultimate 3000 (Thermo Fisher Scientific). The MS data were acquired on a LTQ Velos Mass Spectrometer (Thermo Fisher Scientific) in positive mode with a full MS survey scan over the 400–1600 $m/z$ range and MS/MS scans over the 65–2000 $m/z$ range for the 20 most intense MS ions with a charge of two or more and a collision energy set to 35 eV with Xcalibur$^{TM}$ (v2.2, Thermo Fisher Scientific). Proteins were identified using the MASCOT algorithm (v2.4 Matrix Science) through the Proteome Discoverer$^{TM}$ (v1.4, Thermo Fisher Scientific) software against the Swiss-Prot Human database (downloaded September 2016, 72,285 entries) using a tolerance on the mass measurement of 8 Da in MS mode and 5 Da for MS/MS ions. Results were validated with Mascot Percolator based on a $q$-value < 0.01 and a false discovery rate threshold of 1%.

**Western blot analysis**. Heated protein lysates were separated by 10% SDS-PAGE (Loading buffer 6× (0.35 M Tris HCl pH 6.8, 10.3% SDS, 36% glycerol, 5% β-ME, 0.012% bromophenol blue) and electro-transferred overnight onto polyvinylidene difluoride (PVDF) membranes (GE Healthcare, Cat# 10600023) at 4 °C. PVDF membranes were blocked in Tris-buffered saline containing 0.05% Tween 20 (TBS-T) + 5% milk for 1 h and then incubated overnight with primary antibodies, mouse anti-FLAG (Sigma$^{TM}$, Cat# F1804), rabbit anti-NiV W-CTD (GeneScript$^{TM}$), mouse anti-GAPDH (Chemicon$^{TM}$, Cat# MAB374), rabbit anti-14-3-3 pan (Merck millipore$^{TM}$, Cat# AB9748-I), rabbit anti-NIV N (Valbex, as described[73], anti-NiV W or anti-NiV P/V/W antibodies (GeneScript$^{TM}$ as described[37], mouse anti-HA antibody (Sigma$^{TM}$, Cat#H3663), or anti-NF-κB p65 or p65-S536$^P$, (Cell Signaling, Cat# 8242 S or Cat# 3033 S) diluted 1 : 1000 in TBS-T + 5% milk. Membranes were then washed five times using TBS-T (Euromedex, Cat# ET220-B and VWR, Cat# 28829.296) and incubated for an additional 1 h with horseradish peroxidase-conjugated anti-mouse or anti-rabbit IgG antibodies (Promega$^{TM}$, Cat# W4021 or Cat# W4011) (diluted 1 : 5000 in TBS-T + 1% milk). Membranes were then washed three times in TBS-T, incubated in the Covalight reagent (Covalab$^{TM}$, Cat# opr 0009/00006367) to stain the cell lysates (input) or in the Super Signal West Femto reagent (ThermoScientific$^{TM}$, Cat# 34096) to stain the eluted IPs. Chemiluminescent signals were measured with the VersaDoc Imaging System (Bio-Rad$^{TM}$, http://www.bio-rad.com/webroot/web/pdf/lsr/literature/Bulletin_5609.pdf).

**RNA extraction and quantitative reverse-transcription PCR**. At indicated timepoints, cells were collected and RNA was extracted using the NucleoSpin RNA Kit (Macherey-Nagel, Cat# 740955.250) according to the manufacturer's instructions, and the yield and purity of extracted RNA were assessed using the DS-11-FX spectrophotometer (DeNovix, https://www.denovix.com). Equal amounts of extracted RNA (500 ng) were reverse transcribed using the iScript Select cDNA Synthesis Kit (Bio-Rad, Cat# 170-8891) and amplified by real-time PCR using Platinum SYBR Green qPCR SuperMix-UDG (Invitrogen, Cat# 11744-500) on a StepOnePlus Real-Time PCR System (Applied Biosystems). Data were analyzed using StepOne software (Thermo Fisher Scientific, v2.3, https://www.thermofisher) and calculations were done using the $2^{-\Delta\Delta CT}$ method. Expression was normalized to that of GAPDH (Human GAPDH. For: 5′-CACCCACTCCTCCACCTTTGAC-3′, Rev: 5′-GTCCACCACCCTGTTGCTGTAG-3′) and expressed as copies of mRNA. Specific sets of primers were designed and validated for the detection of human IL-6 (For: 5′-AGAACAGATTTGAGAGTAGTGAGGAAC-3′; Rev: 5′-TTGGGTCAGGGGTGGTTATTGC-3′) and IL-8 (For: 5′-TGCACGGGAGAA-TATACAAATAGC-3′, Rev: 5′-TCTAGCAAACCCATTCAATTCCTG-3′).

**ImageStreamX analysis**. HeLa cells (2.5 × 10$^6$) were seeded in 100 mm Petri dishes and transfected 20 h later with 10 μg of different plasmids (either pCG-FLAG-mCherry (m) or pCG-FLAG-W (accession number AF376747.1)) using JetPrime (Polyplus$^{TM}$, Cat# 114-15) according to the manufacturer's protocol. Four hours post transfection, fresh medium was added and cells were grown for an additional 15 h. Twenty hours post transfection, depending on the condition, cells were stimulated with 10 ng/mL of IL-1β (PeproTech$^{TM}$, Cat# 200-01B) for 20 min. All cells were collected and stained with the Fixable Viability Dye eFluor660 (eBioscience, Cat# 65-0864-14) (1 : 1000) for 15 min. After washing, cells were fixed with 3.7% formaldehyde (Thermo Scientific, Cat# 28908) and permeabilized with PBS (Gibco, Cat# 14190-094) and 0.5% Triton (Sigma Aldrich, Cat# T8787) for 5 min. Cells were incubated for 1 h with primary antibodies as follows: mouse anti-FLAG (Sigma$^{TM}$, Cat# F1804) (1 : 200) or rabbit anti-NF-κB p65 (Santa Cruz, Cat# sc-372) (1 : 100) in 0.1% Triton (Sigma Aldrich, Cat# T8787), 2% FCS (Dutscher$^{TM}$, Cat# S1810-500), 25 mM EDTA (Invitrogen$^{TM}$, Cat#15575-038) in PBS (Gibco, Cat# 14190-094). After three washes with 25 mM EDTA (Invitrogen$^{TM}$, Cat#15575-038), 2% FCS (Dutscher$^{TM}$, Cat# S1810-500) in PBS, cells were incubated for 45 min with secondary antibodies as follows: anti-rabbit conjugate Alexa Fluor 488 (Invitrogen$^{TM}$, Cat# A11034) (1 : 500) and anti-mouse conjugate Alexa Fluor 555 (Invitrogen$^{TM}$, Cat# A31570) (1 : 500) in 0.1% Triton (Sigma Aldrich$^{TM}$, Cat# T8787), 2% FCS (Dutscher$^{TM}$, Cat# S1810-500), 25 mM PTA EDTA

(Invitrogen$^{TM}$, Cat#15575-038) supplemented with DAPI (Thermo Fisher$^{TM}$, Cat# 62248) (1 : 500). Cells were analyzed by ImageStreamX Mark II (Merck Millipore, https://www.luminexcorp.com). Image analysis was performed using the nuclear localization Wizard from the IDEAS Image analysis software (Merck Millipore, https://www.luminexcorp.com). Cell populations were selected hierarchically in the following order: single cells, unsaturated signal, alive, and positively labeled with DAPI. Transfected and non-transfected cells were differentiated. Among the two populations, cells positively labeled for p65 and having an unfragmented nucleus were selected. A pixel-by-pixel similarity score between the DAPI and p65 markings was done to determine the intracellular localization of p65. Cells with a low similarity score do not have a spatial correlation between the two stainings, indicating a predominantly cytoplasmic distribution of p65. In contrast, cells with a high score have a positive correlation between the two stainings, indicating a predominantly nuclear distribution of p65. The percentage of cells with a score higher than 1 (threshold for nuclear localization) was determined for the two cell types. Pictures were taken with a magnification of ×40.

**Immunofluorescence**. For transfection, 3 × 10$^5$ cells were seeded on μ-Slide 8 Wells (Ibidi$^{TM}$, Cat# 80826) and 20 h later transfected with a total of 2 μg of different plasmids using JetPrime (Polyplus$^{TM}$, Cat# 114-15) for HPMECs and Trans IT LT1 (MirusBio, Cat# MIR2300) for HeLa cells, according to the manufacturer's protocol. Four hours post transfection, fresh medium was added and cells were grown for an additional 20 h. Twenty-four hours after transfection, cells were stimulated or not with 10 ng/mL of IL-1β (PeproTech$^{TM}$, Cat# 200-01B) for 20 min before being washed with PBS (Gibco, Cat# 14190-094) and fixed in 3.7% formaldehyde (Thermo Scientific, Cat# 28908) for 20 min. Cells were then permeabilized and blocked using a handmade permeabilization and blocking solution of PBS 3% BSA (Sigma Aldrich, Cat# A3059), 0.15% Triton X-100 (Sigma Aldrich, Cat# T8787). After incubation, cells were immunostained overnight at 4 °C with primary antibodies diluted in permeabilization and blocking solution. After three washes of 3 min each in PBS, cells were incubated for 1 h at RT with the corresponding mix of secondary antibodies containing 1 : 1000 DAPI (Thermo Fisher$^{TM}$, Cat# 62248) to stain the nuclei. Then, cells were washed again in PBS three times for 3–5 min and mounted with Fluoromount-G (SouthernBiotech™, Cat# 0100-01) before observation by confocal microscopy (Zeiss$^{TM}$, LSM800 and LSM980). The primary antibody used were as follows: mouse anti-FLAG (Sigma$^{TM}$, Cat# F1804) (1 : 200), rabbit anti-14-3-3 pan (Merck millipore$^{TM}$, Cat# AB9748-I) (1 : 100), rabbit anti-NF-κB p65 (Cell Signaling, Cat# 8242 S) (1 : 100), or biotinylated rabbit anti-NiV W-CTD (GeneScript$^{TM}$) (1 : 100), and mouse anti-HA (Sigma™, Cat# H9658) (1 : 100). The secondary regents were Alexa 488-conjugated anti-rabbit (Invitrogen$^{TM}$, Cat# A11034), Alexa 555-conjugated anti-mouse (Invitrogen$^{TM}$, Cat# A31570), Alexa 647-conjugated anti-mouse (Invitrogen$^{TM}$, Cat#A31571), Alexa 647-conjugated streptavidin (Invitrogen$^{TM}$, Cat# S21374), and Alexa 555-conjugated streptavidin (Invitrogen$^{TM}$, Cat#S21381) diluted 1 : 500 into the permeabilization and blocking solution.

**Statistics and reproducibility**. Cytokine quantification in the plasma of individual animals was done in duplicate and averaged to derive individual values. Only those cytokines for which a sufficient number of measurements was obtained over the course of infection were retained for analysis. Results are expressed as the mean of concentration, with error bars representing the SD. For each condition, linear regression was performed to assess whether the slope of the fitted model was significantly different from 0 (**$0.001 < p < 0.01$) and the linear model was added to each graph. Normality of residuals was evaluated by a Shapiro–Wilk's test. Details of the statistical analyses are given in Supplementary Table 1.

Analyses of reporter gene assay results were performed for Figs. 1e, f, 2a, d, e, and 4a, and Supplementary Fig. 3b in triplicates (three wells in a plate per condition), and the experiment was repeated at least in three independent experiments. In condition of transfection, each well was normalized by calculating the ratio of the Firefly luciferase activity to the *Renilla* luciferase activity. For a given condition, normal distributions of non-averaged technical replicates across several assays were tested using a Shapiro–Wilk's test. The values for the triplicates of each condition tested on the same plate were then averaged. In addition, condition/positive control ratios of the averaged values were calculated for Figs. 1e, f, 2e, and 4a, and Supplementary Fig. 3b. For Figs. 2a, d and 4a, and Supplementary Fig. 3b, different conditions (stimulated or not) were compared to the control (stimulated or not). For Fig. 1e, different conditions were compared to the non-infected control. For Fig. 1f, different conditions were compared to the infected and stimulated controls. Pairwise comparisons of interest were carried out using the Student's $t$-test if normality was assumed (with correction in case of non-homogeneity of variances (assessed by Fisher's F test)), or otherwise using the Mann–Whitney's rank test. For Fig. 2e, multiple comparisons are carried out using one-way analysis of variance (ANOVA), completed by a Tukey's multiple comparisons test (*$p < 0.05$, **$p < 0.01$, ***$p < 0.001$, and ****$p < 0.0001$).

Densitometric analysis of phosphorylated p65 immunoblots from three independent experiments was performed using the VersaDoc Imaging System (Bio-Rad, http://www.bio-rad.com/webroot/web/pdf/lsr/literature/Bulletin_5609.pdf) and analyzed with ImageJ 1.52p Fiji package software (https://imagej.net/Fiji). GAPDH expression was used for normalization.

Quantitative reverse-transcription PCR analysis of cytokine production is presented in the form of histograms that show the mean number of mRNA copies of a gene divided by the mean number of mRNA copies of the control (Ø) for each condition (values expressed as fold changes). Error bars represent the SD for $n = 3$ experimental replicates. Different conditions were compared to the control (Ø). Statistical significance was assessed by one-way ANOVA completed with a multiple comparisons Tukey's test (*$p < 0.05$, **$p < 0.01$, ***$p < 0.001$, ****$p < 0.0001$).

ImageStreamX results are presented in the form of histograms that represent the mean percentage of cells that have NF-κB p65 localized in the nucleus and error bars represent the SD for $n = 3$ experimental replicates. Different conditions were compared to the control (Ø). Statistical significance was assessed by one-way ANOVA and a multiple comparisons Tukey's test (*$p < 0.05$).

For microscopy analysis a plot profile was performed on raw pictures obtained by confocal microscopy (Zeiss™ LSM800) and data were exported to GraphPad Prism 8.3.0 (GraphPad Software, Inc., https://www.graphpad.com/scientific-software/prism) for graphical representation and statistical analysis. Spectra were obtained using the ImageJ 1.52p FiJi package. For each condition, the shortest distance between each part of analyzed cells was taken to determine the fluctuation of endogenous cellular proteins inside the nucleus and the basal level of expression of the proteins of interest in the cytoplasm. Each experiment was done two to three times usually in triplicate or duplicate (as stated accordingly) and nine cells taken from nine different fields were selected randomly by the device or by the user. Shown quantification of nuclear localization of 14-3-3 or p65 are from a single representative analysis of randomly selected pictures of cells taken within the same confocal analysis session to minimize any possible technical induced variation between imaging sessions. For the quantification reported in Figs. 4–6 and Supplementary Figs. 6, 7, each dot represents the mean fluorescence intensity for 14-3-3 staining in the nuclei of transfected cells, divided by the mean fluorescence in non-transfected cells in the same well (values are expressed as fold change) and error bars represent the confidence interval of the mean (CI 95%) for $n = 9$ cells per condition. The normal distributions of the values for each condition were tested using a Shapiro–Wilk's test. Different conditions were compared to the control (Ø). Statistical significance was assessed by a one-way ANOVA and multiple comparisons Tukey's test for Figs. 4d and 6b, and by a nonparametric Kruskal–Wallis test with Dunn's multiple comparisons for Fig. 5 (*$p < 0.05$, **$p < 0.01$, ***$p < 0.001$, ****$p < 0.0001$). For Fig. 6d, each dot represents the mean fluorescence intensity for the 14-3-3 protein in infected cells divided by the mean fluorescence in non-infected cells in the same section for a given animal (values expressed as fold change) and error bars represent the CI 95% for $n = 5$ sections (4–9 cells) per condition. The normal distributions of the values were tested using a Shapiro–Wilk's test. The infected animal was compared to the non-infected control. Statistical significance was assessed by a t-test (***$p < 0.001$).

**Reporting summary**. Further information on research design is available in the Nature Research Reporting Summary linked to this article.

## Data availability
The authors declare that the data supporting the findings of this study are available from the corresponding author upon request. All relevant data including the numerical and statistical source data that underlie the graphs in figures are provided in Supplementary Data 1.

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

## Acknowledgements

The work was supported by INSERM, LABEX ECOFECT (ANR-11-LABX-0048) of Lyon University, within the program "Investissements d'Avenir" (ANR-11-IDEX-0007) operated by the French National Research Agency (ANR), by ANR-18-CE11-0014-02, and by Aviesan Sino-French agreement on Nipah virus study. We thank the animal experimentation team of Inserm "Jean Mérieux" BSL4 laboratory for the realization of the animal experiment and the biosafety team for their assistance for BSL4 activities. We are also indebted to Sophie Shyfrin, a native English speaker, for having carefully edited our manuscript. We are grateful to O. Reynard, K. Dhondt, Q. Bardin, A. Linder, and all the members of the group Immunobiology of viral infection at CIRI, and S. Reynard, UBIVE-CIRI, Lyon, for the help in the realization of this study, and Pierre E. Rollin and the Center for Disease Control and Prevention, Atlanta USA, for providing the Nipah virus Bangladesh isolate. We acknowledge the contribution of the SFR Biosciences (UMS3444/CNRS, US8/Inserm, ENS de Lyon, UCBL) facility Lymic-Platim-Microscopy (J. Brocard and E. Chatre).

## Author contributions

F.E., M.I., B.H., L.M.B., C.J. and D.G. designed the study. F.E., C.D., N.A., M.I., R.P., L.M.B., and C.M. performed experiments. F.E., B.H., M.I., C.D., M.I., R.P., D.G., C.M., and C.J. analyzed the data. F.E., B.H., and D.G. wrote the article. F.E., M.I., C.D., B.H., R.P., and D.G. prepared the figures. C.J. and C.C. provided some essential tools.

## Competing interests

The authors declare no competing interests.
