## [Transparent Peer Review File · Communications Biology]

Reviewers' comments:

Reviewer #1 (Remarks to the Author):

This manuscript by Enchery et al., described studies focused on determining the role of the Nipah virus (NiV) W protein in regulation of the NF- κ B driven inflammatory response during NiV infection. The authors found that the W protein binds to the molecule 14-3-3 in the nucleus and inhibits NF- κ B signaling. The studies described here show that W protein inhibits TNF α and IL-1 β induced NF- κ B activity and appears to marginally decrease the accumulation of p65 in the nucleus. Further data indicates that W binds 14-3-3 and that a mutation at Serine 449 in the C-terminal domain (CTD) of W may play a role in reducing the NF- κ B response, presumably by serving as a binding site for 14-3-3. The authors test their hypothesis in HeLa cells and pulmonary epithelial cells with similar results. The author also used tissue collected from a study with African green monkeys and present data suggesting that W accumulates in lung cell nuclei along with 14-3-3. The authors also present a model showing their hypothesis regarding the mechanism of action proposed for W-based inhibition of NF- κ B activity.

In general, the data presented in this manuscript supports the proposed hypothesis. However, this manuscript feels like a data dump of for all of the work completed on this project rather than focusing on the most salient experiments that prove the proposed hypothesis. Further, there are fundamental aspects of this manuscript that are not well explained and a document that is already data dense, even more difficult to understand. Some fairly basic improvements could make this submission much easier to digest.

Specific comments:

Figure 1: A critical aspect of this paper is the work performed with the W CTD. Unfortunately, what is the W CTD is not clearly explained. Digging through the methods answers the question, but it would be helpful if the authors improved figure 1 to make it clear which constructs are being tested and what they are called. Further, the AGM studies were performed with the Bangladesh isolate which can cause a different disease in this species than the Malaysian isolate. Presumably the W constructs were generated using the same isolate. Are there sequence differences between the Malaysian and Bangladesh isolate in the W CTD that could impact pathogenicity/regulation of NF- κ B? This should be stated.

Page 3, Line 15: Along with the above, W-CTD is not defined. Nor is CTD (line 36) or NTD in general.

Page 4, line 6 W-CTD and W-NTD data for TNF α stimulation is not shown

Page 4, line 19: These series of mutants should be in Figure 1a. They aren't. Personally, I would expand the W protein part of the figure to highlight the mutants used in this study.

Page 4, line 42: Is this meant to be 14-3-3(theta)?

Page 5, line 13: This line is not clear.

Page 5, line 19: This bit is fairly important as it plays a role in your overall hypothesis. In my first reading I missed that these were predicted phosphorylation sites. Do you have data to show that the W protein is phosphorylated on these sites? You have IP assays that could be used to help demonstrate serine phosphorylation by probing alanine variants vs WT. This should be done to support your model. I appreciate that the alanine mutation removes the putative phosphorylation site, but...

Page 5, line 31: Please state the two cell types.

Page 6, line 4: The image in figure 5a looks more nuclear to me, I would even suggest bound to the nuclear membrane.

Page 8, lines 11-26: Some of this is redundant with the previous paragraph. You might consider consolidating.

Figure 3C: Was this probed with an anti-W Ab? Legend doesn't indicate this is the case and it seems like it should. I don't understand this figure otherwise.

Figure 4: What was the rationale for using a 20h transfection for NF- κ B expression and 48h for p65? It seems to be testing at the same time point would more clearly demonstrate a direct association.

Figure 5d: I don't suppose you still have this section? All of your other images are very good, this one is quite blurry. I am guessing this was enhanced in imaging software rather than using a higher magnification.

Reviewer #2 (Remarks to the Author):

The manuscript from Enchéry et al. addresses an important and significant area of the virus-host interface, specifically immune evasion/subversion. The research includes analysis of virus infection of non-human primates with the highly pathogenic (BSL-4) pathogen, Nipah virus, to identify effects on cytokines likely to be produced by NF κ B in virus infected cells, and a series of in vitro studies to elucidate the mechanism underlying viral subversion of NF κ B-dependent signaling. Using transfection expression of W protein and variants thereof, and infection by recombinant virus lacking the W protein unique C-terminal domain indicate that the function relates to this region. The paper further proposes that interaction of W protein with 14-3-3 underpins this function.

The findings are of clear interest in the field. The impact of some of the findings is dampened by recent publication (appropriately cited in this paper) that W protein interacts with all isoforms of 14-3-3 via the C-terminal domain, and identifying critical serine residue, and demonstrating that this impacts transcription in response to infection by a heterologous virus.

However, the current submission proposes a mechanism whereby nuclear localization of 14-3-3 by the nuclear W protein (dependent on NLS-driven nuclear localization of the latter) exploits cellular regulatory mechanisms to suppress NF κ B signaling. This is very interesting as a model as it contributes to knowledge of immune evasion and of the role of nuclear trafficking of viral proteins in a cytoplasmic virus.

A weakness in this is that the model relies largely on observed effects that appear to correlate, but could be coincidental; i.e. it is entirely possible that W/CTD etc. effect NF κ B by a mechanism independent of interaction with 14-3-3. Some data directly supporting dependence of the effect on 14-3-3 would greatly enhance the paper – I realise this may not be straightforward due to the nature of 14-3-3 isoforms.

I also consider that additional discussion of the nature of the immune-pathology indicated in the paper, and how this might relate to the inflammatory gene profiles observed here and in the previous paper would be valuable. It is also not entirely clear how the model proposed accounts for apparent effects on NF κ B phosphorylation; some discussion on this point would be valuable.

Finally, some of the confocal data for p65 requires additional consideration to be entirely convincing of the interpretation.

Major points/queries on data:

1. Is it possible to examine directly the dependence of the effect of W protein on NF κ B on 14-3-3. Is it possible to knock-out or knock-down predominant 14-3-3 isoforms, or to use inhibitors of 14-3-3?

Demonstrating that the effect of W protein has a significant dependence on 14-3-3 would directly support the proposed model. Based on the model in Fig. 6, I also wondered whether one might predict greater interaction of 14-3-3 with p65 following wt but not defective mutant W expression.

2. Some additional exposition would be useful on how the proposed model accounts for effects on phosphorylation of p65 (does this fit current knowledge on 14-3-3 and NFkB), and also on how the observed effects on gene expression might account for known inflammatory/immune disease outcomes.

3. Extended data fig 1 is cited (In 5 page 4) as showing viability/transfection efficiency for all conditions – I don't think it does (e.g. different variants, non-treated/treated etc. are missing?).

4. Following point 3, the expression of proteins/cell viability etc. is something I would query generally – it appears that for luciferase assays a transfection control (renilla) is included, which is good and might be highlighted in the legend/text as it indicates that altered expression is specific and does not relate to viability, general effects on transcription etc. However, there is a lack of any clear evidence that the altered effects of W protein variants is independent of altered expression of the transfected proteins. Inclusion of westerns of lysates corresponding to signaling assays would be very valuable to support specific effects.

5. Pg 4 “possibly because of incomplete inhibition of nuclear import of this variant (Fig. 2f)” – can this be shown in this cell type?

6. The MS list cited in the text would be a very valuable resource – will this be provided in the final manuscript?

7. In confocal data, it seems only some images are shown; for example: “These phenotypes, observed both with and without stimulation of cells by IL-1 β (Fig. 5a, b, Extended Data Fig. 5),” – are images without stimulation shown? Also data for some confocal appears to use different cell types – can this be discussed?

8. Figure 1f – is there data for cell viability/viral protein expression etc. for treated cells, to support specificity of the difference observed?

9. Figure 2 and other figures testing multiple conditions/W variants for signaling – are there any data for protein expression (e.g. western blot of lysates for viral proteins and loading control); if a control renilla construct was used here, should mention in the figure legend with the list of transfections.

10. The data often indicate that W CTD alone is more potent than WT W protein – can the authors provide discussion of this point.

11. In confocal data Fig 4 p65 localization is highly variable (heterogeneous) – i.e. some cells show cytoplasmic localization irrespective of expression of W protein; conversely, in S449A image, nuclear localization of p65 is only apparent in a cell with very dim red staining (W protein) while in at least 2 other cells with clear high expression of W protein, p65 is excluded. It is difficult to judge these data, and additional data or quantification across multiple cells would be required to be convincing. Analysis of individual cells shown is not necessarily representative, given the clear cell-cell variation.

12. In confocal data for Fig 5, there is quantification, but this is of a relatively low number of cells and it is normalised to a low number of non-transfected cells. I do not quite understand the analysis resulting in all cells in empty vector or nt being equal to 1. If each individual cell is calculated relative to a mean of non-expressing cells in the same well (methods), then surely there should be some variation, which would be accounted for in the statistical analysis. Clear definition of how the numbers were derived should be included – a value of 1 indicates each cell is divided only by itself, which will not account for variation in the sample.

Other points:

1. Suggest short hand use of 'W' could be modified to 'W protein'

2. A careful proof read is important – in some areas, meaning is unclear: e.g. signaling, restoring the production of W-mediated inhibition of proinflammatory cytokines; induces nuclear accumulation of 14-3-3 both in vitro and in vivo dampens the production; help explaining why Henipaviruses are so particular among the Paramyxoviridae family; wild type or recombinant NiV expressing or not W protein truncated from its CTD.

3. Perforin is spelled perforine in figure
4. Figure 1 western – can position of all isoforms be labelled; can expression level of isoforms be estimated by densitometry?
5. Line 7 page 4 – the idea that W/NTD has ‘no effect’ is not entirely clear to me – it appears to not differ significantly from the control, but also may not differ from WT.
6. Page 4: “Altogether, these results suggest that NF- κ B inhibitory effect of W occurs downstream IKK β , but cannot counteract the overexpression of p65.”; also pg. 7 “Furthermore, this could explain the inability of W to inhibit the transcriptional activation by overexpressed p65 (Fig. 2d).” – can this be expanded on to clarify why this is expected, particularly if W acts at the level of 14-3-3 interaction with p65.
8. Use of ‘quoted’ – this should probably be ‘indicated’ or ‘denoted’ depending on the context.
9. Fig 2f – the statistics should probably be used to show whether the NLS0 version of W differs from the WT (this is the key data?)
10. Ln36, P4 “W interacts with endogenous 14-3-3 via its CTD S449 residue in” should be “dependent on”?
11. Ln9, P4 – somehow should perhaps be somewhat
12. Some of the data in the extended figures does not appear to match the text – e.g. “In contrast, the expression of either W-S449A, W-S199A-S449A or W NTD did not affect the IL-1b-induced p65 intranuclear translocation (Fig. 4c, Extended Data Fig. 3)”.
13. “14-3-3 was distributed equally in the cytoplasm and nucleus of the cell (Fig. 5a)” – it appears more nuclear than cytoplasmic to me
14. Page 6 – references to Extended Data appear incorrect
15. Pg 7: “14-3-3 can force the nuclear export of the ligand by enabling the function of its NLS” – is this correct?
16. In discussion, a brief mention of the mechanism of NF κ B deactivation/export with I κ Ba would be valuable.
17. (Rodriguez, Cruz and Horvath 15 2004) reference in methods should be formattedz
18. In Methods, Analysis of the results form reporter gene assay results could be clearer – e.g. is the normality tested using non-averaged technical replicates across several assays; do all assays compare multiple biological replicates etc. Some areas of the text are not wholly clear.
19. “RTqPCR analysis of cytokines production” should be bold (title)
20. Figure 2d description in legend is not clear.
21. Figure 3b – can a control for specificity of IP (e.g. probe for GAPDH) be included to confirm detection of W in IP from infected cells is specific?
22. Although the IP input for HPMEC supports the findings, can a clearer input for HeLa be included?
23. Fig 5 – why were treated (IL1b) and untreated cells analysed – is there an effect of treatment, and can non-treated cell images be shown?
24. Fig 5d – Images for infected cells are out of focus/unclear in the pdf.; the zoom appears to select a particularly bright green area; lines to the zoom are incomplete
25. Extended Fig 3 – is this the intended figure, and why is p65 cytoplasmic in IL1b-treated nt cells? Why is there no p65 in some cells – is this phospho-p65 (and if so, why is it still excluded from the nucleus in nt cells + IL1b)?
26. Extended fig 5; some of the cells appear quite different to expected (e.g. some cells appear to exclude 14-3-3 in W-NTD) – there almost appears to be some colocalization with W-NTD – can the authors comment on this.
27. Extended table – requires some revision – it is not clear what +/- indicates, and several comments appear incorrect (e.g. Nuclear import of NiV W is dispensable for NF- κ B inhibition)

Rebuttal Letter

Reviewer #1 (Remarks to the Author):

This manuscript by Enchery et al., described studies focused on determining the role of the Nipah virus (NiV) W protein in regulation of the NF- κ B driven inflammatory response during NiV infection. The authors found that the W protein binds to the molecule 14-3-3 in the nucleus and inhibits NF- κ B signaling. The studies described here show that W protein inhibits TNF α and IL-1 β induced NF- κ B activity and appears to marginally decrease the accumulation of p65 in the nucleus. Further data indicates that W binds 14-3-3 and that a mutation at Serine 449 in the C-terminal domain (CTD) of W may play a role in reducing the NF- κ B response, presumably by serving as a binding site for 14-3-3. The authors test their hypothesis in HeLa cells and pulmonary epithelial cells with similar results. The author also used tissue collected from a study with African green monkeys and present data suggesting that W accumulates in lung cell nuclei along with 14-3-3. The authors also present a model showing their hypothesis regarding the mechanism of action proposed for W-based inhibition of NF- κ B activity.

In general, the data presented in this manuscript supports the proposed hypothesis. However, this manuscript feels like a data dump of for all of the work completed on this project rather than focusing on the most salient experiments that prove the proposed hypothesis. Further, there are fundamental aspects of this manuscript that are not well explained and a document that is already data dense, even more difficult to understand. Some fairly basic improvements could make this submission much easier to digest.

We thank reviewer for the constructive critics which we tried to address in the revised manuscript. We have performed the additional experiments and included them in the manuscript and we provide below a point-by-point answers the points raised by the referee.

Specific comments:

Figure 1: A critical aspect of this paper is the work performed with the W CTD. Unfortunately, what is the W CTD is not clearly explained. Digging through the methods answers the question, but it would be helpful if the authors improved figure 1 to make it clear which constructs are being tested and what they are called. Further, the AGM studies were performed with the Bangladesh isolate which can cause a different disease in this species than the Malaysian isolate. Presumably the W constructs were generated using the same isolate. Are there sequence differences between the Malaysian and Bangladesh isolate in the W CTD that could impact pathogenicity/regulation of NF- κ B? This should be stated.

A clear definition of W NTD and CTD has been included in the body of the text of the revised manuscript (P2 lines 41-42), and the Figure 1a was enriched with the delineation of the different domains and localization of the variants used in the different experiments. The amino acid sequences of W CTD from NiV-M and NiV-B isolates are identical and their NTD share 87% identity as now shown the Supplementary Fig. 1 and mentioned additionally at P2, lines 42-43. Therefore, we expected identical impact of NiV W protein from two NiV strains on NF- κ B regulation.

Fig 1a, Schematic representation of the NiV genome and the proteins encoded by the P gene: structural P and non-structural proteins, C (alternative initiation codon), V and W, the latter two resulting from P mRNA editing. Residues important for the STAT1/4-binding domain, NES and NLS within the W protein are indicated. Le, Leader; Tr, Trailer.

Supplementary Fig. 1. Comparison of NiV W sequences between NiV Malaysia (NiV-M, AF212302) and NiV Bangladesh (NiV-B, (NiV-B, AY988601) isolates. W NTD and CTD of the two isolates share 87% and 100% identity respectively. Residues important for the STAT 1 and 4-binding domain, NES and NLS within the W protein are indicated. Red squares correspond to the RXX(Ser/Thr)XP motif presenting a potential site of interaction with 14-3-3.

Page 3, Line 15: Along with the above, W-CTD is not defined. Nor is CTD (line 36) or NTD in general.

As describe above, the clear definition of W NTD and CTD has been included in the body of the text of the revised manuscript (P2, lines 40-43)

Page 4, line 6 W-CTD and W-NTD data for TNFa stimulation is not shown.

We thank reviewer for pointing that omission. Indeed, only results from IL-1b stimulation in the presence of W-NTD & CTD were presented in the figure 2a and this has been corrected in the revised manuscript (P4, lines 6-7).

Page 4, line 19: These series of mutants should be in Figure 1a. They aren't. Personally, I would expand the W protein part of the figure to highlight the mutants used in this study.

Details of W variants are now indicated in revised Figure 1a.

Page 4, line 42: Is this meant to be 14-3-3(theta)?

Yes, "θ" is the Greek symbol for "theta" (as "μ" is for "mu").

Page 5, line 13: This line is not clear.

The sentence was modified for more clarity (P5, lines 9-13).

Page 5, line 19: This bit is fairly important as it plays a role in your overall hypothesis. In my first reading I missed that these were predicted phosphorylation sites. Do you have data to show that the W protein is phosphorylated on these sites? You have IP assays that could be used to help demonstrate serine phosphorylation by probing alanine variants vs WT. This should be done to support your model. I appreciate that the alanine mutation removes the putative phosphorylation site, but...

As suggested by the reviewer, we have performed the additional experiments to demonstrate the importance of the phosphorylation of NiV W for the interaction with 14-3-3. As presented in the new Supplementary Fig. 4 (P5, lines 27-32). HPMEC cells were transfected with a plasmid encoding Flag-tagged W or truncated/variant W proteins and cellular extracts were treated or not with Lambda phosphatase prior to immunoprecipitation. Co-immunoprecipitation of exogenous Flag-W proteins with endogenous 14-3-3 proteins was performed using anti-Flag antibodies and input cell extracts were analyzed by western blot using anti-FLAG, anti-14-3-3 and anti-GAPDH antibodies. The obtained results demonstrate clearly that following the treatment with lambda phosphatase that removes all protein phosphorylation, the interaction between W and 14-3-3 is lost, demonstrating that NiV-W phosphorylation is primordial for the interaction with 14-3-3, in agreement with the very recently reported crystal structure of W CTD peptide-14.3.3 complex describing that phosphate group on W-S449 could interact with the bonding groove of 14-3-3 proteins (Edwards et al, Ref N°41).

Supplementary Figure 4: Phosphorylation at S449 on W-CTD protein is responsible for the interaction with 14-3-3. HPMEC cells were transfected with a plasmid encoding Flag-tagged W or truncated/variant W proteins; an empty vector (\emptyset) was used as control. Cellular extracts were treated or not with Lambda phosphatase prior to immunoprecipitation. Co-immunoprecipitation of exogenous Flag-W proteins with endogenous 14-3-3 proteins using anti-Flag antibodies bound to agarose beads and detected by western-blot

(IB) using rabbit anti-pan 14-3-3 antibodies. Input cell extracts were analyzed by western blot using anti-FLAG, anti-14-3-3 and anti-GAPDH antibodies.

Page 5, line 31: Please state the two cell types.

Modifications were done in the revised manuscript. 4a = HPMEC & extended 2b = HeLa (P5, line 38).

Page 6, line 4: The image in figure 5a looks more nuclear to me, I would even suggest bound to the nuclear membrane.

Figure 5a was modified with enlarged images from individual cells allowing to better appreciate the intracellular localization of 14-3-3 and the statistical analysis was included (see below).

Page 8, lines 11-26: Some of this is redundant with the previous paragraph. You might consider consolidating.

Modifications were done in the last paragraph of the discussion of the revised manuscript.

Figure 3C: Was this probed with an anti-W Ab? Legend doesn't indicate this is the case and it seems like it should. I don't understand this figure otherwise.

It was probed with an anti-Flag Ab, as we used Flag-tagged W constructs. Legends were modified in figure 3c as in figure 3a to avoid confusion.

Figure 4: What was the rationale for using a 20h transfection for NF-kB expression and 48h for p65? It seems to be testing at the same time point would more clearly demonstrate a direct association.

The high sensitivity of the luciferase activity allows measurement of the NF-kB activation through normalization with Renilla while detection of the phosphorylation of p65 through immunofluorescence is better detected at 48h post transfection, time period allowing optimal expression of W protein.

Figure 5d: I don't suppose you still have this section? All of you other images are very good, this one is quite blurry. I am guessing this was enhanced in imaging software rather than using a higher magnification.

Images in the figure 5d (5C in the revised manuscript) are now replaced by images of better quality, presented below.

Reviewer #2 (Remarks to the Author):

The manuscript from Enchéry et al. addresses an important and significant area of the virus-host interface, specifically immune evasion/subversion. The research includes analysis of virus infection of non-human primates with the highly pathogenic (BSL-4) pathogen, Nipah virus, to identify effects on cytokines likely to be produced by NFkB in virus infected cells, and a series of in vitro studies to elucidate the mechanism underlying viral subversion of NFkB-dependent signaling. Using transfection expression of W protein and variants thereof, and infection by recombinant virus lacking the W protein unique C-terminal domain indicate that the function relates to this region. The paper further proposes that interaction of W protein with 14-3-3 underpins this function.

The findings are of clear interest in the field. The impact of some of the findings is dampened by recent publication (appropriately cited in this paper) that W protein interacts with all isoforms of 14-3-3 via the C-terminal domain, and identifying critical serine residue, and demonstrating that this impacts transcription in response to infection by a heterologous virus.

However, the current submission proposes a mechanism whereby nuclear localization of 14-3-3 by the nuclear W protein (dependent on NLS-driven nuclear localization of the latter) exploits cellular regulatory mechanisms to suppress NFkB signaling. This is very interesting as a model as it contributes to knowledge of immune evasion and of the role of nuclear trafficking of viral proteins in a cytoplasmic virus.

A weakness in this is that the model relies largely on observed effects that appear to correlate, but could be coincidental; i.e. it is entirely possible that W/CTD etc. effect NFkB by a mechanism independent of interaction with 14-3-3. Some data directly supporting dependence of the effect on 14-3-3 would greatly enhance the paper – I realise this may not be straightforward due to the nature of 14-3-3 isoforms.

I also consider that additional discussion of the nature of the immune-pathology indicated in the paper, and how this might relate to the inflammatory gene profiles observed here and in the previous paper would be valuable. It is also not entirely clear how the model proposed accounts for apparent effects on NFkB phosphorylation; some discussion on this point would be valuable.

Finally, some of the confocal data for p65 requires additional consideration to be entirely convincing of the interpretation.

We thank reviewer for the constructive critics which we tried to address in the revised manuscript. We have performed the additional experiments and included them in the manuscript and we provide below a point-by-point answers the points raised by the referee.

Major points/queries on data:

1. *Is it possible to examine directly the dependence of the effect of W protein on NFkB on 14-3-3. Is it possible to knock-out or knock-down predominant 14-3-3 isoforms, or to use inhibitors of 14-3-3? Demonstrating that the effect of W protein has a significant*

dependence on 14-3-3 would directly support the proposed model. Based on the model in Fig. 6, I also wondered whether one might predict greater interaction of 14-3-3 with p65 following wt but not defective mutant W expression.

We agree with the reviewer that the demonstration of the 14-3-3-dependent effect of W protein is important for the confirmation of the proposed model. However, as 14-3-3 exists in 7 different isoforms, we did not consider the knock-down of 14-3-3 as the best approach to address that question. We have thus generated the plasmid coding for the R18 peptide, demonstrated to inhibit the activity of 14-3-3 (Wang B et al, Biochemistry 1999, Ref N°71). The expression of R18 peptide in HPMEC cells transfected with W and its truncated/variant forms disabled the W-induced inhibition of IL-1 β -induced NF- κ B nuclear translocalisation (**Supplementary Fig 6**), suggesting the role of 14-3-3 in this inhibition, thus comforting the proposed model.

Supplementary Fig 6. Expression of 14-3-3 inhibitory peptide R18 disables W protein-induced inhibition of NF- κ B p65 nuclear localization. HPMEC cells were transfected or not (\emptyset) with R18-HA in addition to indicated W plasmids and its truncated/variant forms and stimulated with 10 ng/ml of IL-1 β for 20 min. Cells were fixed and stained for NF- κ B p65, anti-HA (for staining of R18

flagged with a HA tag) and NiV W and analyzed by confocal microscopy. Representative images of cells transfected with indicated plasmids were presented (scale bar = 10 μ m).

2. Some additional exposition would be useful on how the proposed model accounts for effects on phosphorylation of p65 (does this fit current knowledge on 14-3-3 and NF-kB), and also on how the observed effects on gene expression might account for known inflammatory/immune disease outcomes.

As p65 phosphorylation is considered to be a hallmark of the p65 activation (reviewed in Christian F et al, 2016, ref N°4), we analyzed the effect of W protein and its mutants on p65 phosphorylation on its serine S536 and presented the data in the Figure 5B. Decreased levels of activated p-p65 have the direct effects on the expression of NF-kB-dependent inflammatory process and production of proinflammatory cytokines. More particularly, the S536 phosphorylation was shown to lead to IL-8 transcription (Sasaki C.Y. et al, JBC, 2005, Ref N°45). This has been additionally discussed in the revised manuscript (P5, line 41).

3. Extended data fig 1 is cited (In 5 page 4) as showing viability/transfection efficiency for all conditions – I don't think it does (e.g. different variants, non-treated/treated etc. are missing?).

Indeed, the transfection efficiency and cell viability presented in the extended data Figure 1 corresponds to the conditions used in ImageStremX analysis, presented in the Figure 2B, which has been corrected in the revised manuscript.

4. Following point 3, the expression of proteins/cell viability etc. is something I would query generally – it appears that for luciferase assays a transfection control (renilla) is included, which is good and might be highlighted in the legend/text as it indicates that altered expression is specific and does not relate to viability, general effects on transcription etc. However, there is a lack of any clear evidence that the altered effects of W protein variants is independent of altered expression of the transfected proteins. Inclusion of westerns of lysates corresponding to signaling assays would be very valuable to support specific effects.

A Renilla transfection control is indeed included in luciferase assays and was highlighted in legends of figures in revised manuscript. As the activity of many promoters was shown also to respond directly or indirectly to NF-kB signaling (e.g. CMV promoter (He and Weber 2004), we use a Renilla plasmid without promoter (pRL-Null, Promega), which in our calibration experiments presented a detectable level of Renilla expression compared to not transfected cells. In addition, representative results of Western blot analysis of cell lysates are also presented in the Fig 3.

5. Pg 4 “possibly because of incomplete inhibition of nuclear import of this variant (Fig. 2f)” – can this be shown in this cell type?

Since the other reviewer pointed that our manuscript is overloaded with data presenting the similar results only obtained in different cell lines, we removed figure 2f that presents another cell type (A549 cells).

6. The MS list cited in the text would be a very valuable resource – will this be provided in the final manuscript?

The MS results were obtained from a single experiment performed during this study and obtained data contained most of the proteins which have been already reported. We thus considered that the short list provided in the manuscript is sufficient and that enlarging of the list with non-reported proteins will need an additional experiment for confirmation.

7. *In confocal data, it seems only some images are shown; for example: “These phenotypes, observed both with and without stimulation of cells by IL-1 β (Fig. 5a, b, Extended Data Fig. 5),” – are images without stimulation shown? Also data for some confocal appears to use different cell types – can this be discussed?*

In the figure 5a, only images presenting the conditions with IL-1 β are presented. However, both experiments (with or without stimulation) were performed and results of fluorescence intensity obtained without stimulation are presented in the Extended figure 5 to support the sentence “These phenotypes, observed both with and without stimulation of cells by IL-1 β ” in the text.

Figure 4 presents data obtained using human microvascular pulmonary endothelial cell line HPMEC, as endothelial cells are known to be the target of Nipah virus infection (as stated on page 5 lines 12 of the revised manuscript). Figure 5 used HPMEC and lung sections from monkeys (*in vivo* investigations). Supplementary figures 3 and 9 present data obtained with HeLa cells, to further confirm results on an additional cell type and extend the message with the additional conditions.

8. *Figure 1f – is there data for cell viability/viral protein expression etc. for treated cells, to support specificity of the difference observed?*

The WB presented in figure 1d represents expression of proteins directly collected from a representative session of the functional test presented in figure 1f, including the viral proteins following the infection with indicated viruses. Cell viability was qualitatively evaluated by the relative expression levels of GAPDH.

9. *Figure 2 and other figures testing multiple conditions/W variants for signaling – are there any data for protein expression (e.g. western blot of lysates for viral proteins and loading control); if a control renilla construct was used here, should mention in the figure legend with the list of transfections.*

This point has been answered within the answer to the question 4: the transfection with Renilla was included in luciferase assays. In addition, representative data of Western blot analysis of cell lysates are presented in the Fig 3 to illustrate the controls that were systematically included. The legends of figures were modified accordingly.

10. *The data often indicate that W CTD alone is more potent than WT W protein – can the authors provide discussion of this point.*

The small molecular weight of our W-CTD constructs linked to mCherry (around 35 kDa, in contrast to 70 kDa for W) may facilitate the entry of W-CTD into the nucleus, increasing thus its effect on the NF- κ B activity. This has been additionally mentioned in the revised manuscript (P4, lines 9-10).

In confocal data Fig 4 p65 localization is highly variable (heterogeneous) – i.e. some cells show cytoplasmic localization irrespective of expression of W protein; conversely, in S449A image, nuclear localization of p65 is only apparent in a cell with very dim red staining (W protein) while in at least 2 other cells with clear high expression of W protein, p65 is excluded. It is difficult to judge these data, and additional data or quantification across multiple cells would be required to be convincing. Analysis of individual cells shown is not necessarily representative, given the clear cell-cell variation.

Figure 4c has been modified with enlarged images of cells included for the easier evaluation of the p65 nuclear localization and the statistical analysis performed for each condition (presented below).

Fig. 4 .../.... (c) HPMEC cells were transfected with indicated plasmids and stimulated with 10 ng/ml of IL-1 β for 20 min. Cells were fixed and stained for NF- κ B p65 and NIV W and analyzed by confocal microscopy (scale bar = 10 μ m). (d) The mean fluorescence intensity for p65 protein was measured in cell nucleus. Signal obtained from anti-W labelled cells was normalized to the signal from unlabeled cells and expressed in fold change. Error bars represent the mean's confidence interval (CI 95%) for 9 cells. Statistical significance was assessed by a one-way unpaired ANOVA, multiple comparisons Tukey's test; * p < 0.05, *** p < 0.001 and **** p < 0.0001.

11. In confocal data for Fig 5, there is quantification, but this is of a relatively low number of cells and it is normalised to a low number of non-transfected cells. I do not quite understand the analysis resulting in all cells in empty vector or nt being equal to 1. If each individual cell is calculated relative to a mean of non-expressing cells in the same well (methods), then surely there should be some variation, which would be accounted for in the statistical analysis. Clear definition of how the numbers were derived should be included – a value of 1 indicates each cell is divided only by itself, which will not account for variation in the sample .

The method of the quantification was revised and presented in the figure legend and methods section. Signal

obtained from anti-W labelled cells was normalized with the signal from unlabeled cells and expressed in fold change. Error bars represent the mean's confidence interval (CI 95%) from nine cells. Figure 5 has been additionally modified with enlarged images of cells to allow better appreciation of the nuclear localization of 14-3-3 (as presented above in the response to the reviewer 1).

Other points:

1. Suggest short hand use of 'W' could be modified to 'W protein'.

Modifications were done throughout the revised manuscript.

2. A careful proof read is important – in some areas, meaning is unclear: e.g. signaling, restoring the production of W-mediated inhibition of proinflammatory cytokines; induces nuclear accumulation of 14-3-3 both in vitro and in vivo dampens the production; help explaining why Henipaviruses are so particular among the Paramyxoviridae family; wild type or recombinant NiV expressing or not W protein truncated from its CTD.

The article was proofread and modified according to reviewers' comments.

3. Perforin is spelled perforine in figure 4.

Modifications were done accordingly.

4. Figure 1 western – can position of all isoforms be labelled; can expression level of isoforms be estimated by densitometry?

Positions of all isoforms were labelled in the figure 1d. The relative expression level of N, P, V/W NTD and W CTD were estimated by densitometry. The graph bellow presents the relative expression of the viral proteins in the figure 1d. For the conditions rNiV and rNiVΔWCTD, the densitometry values of viral proteins were normalized with the densitometry value of the GAPDH, taking into account:

- for N, the staining with α -N around 55 kDa,
- for P, the staining with α -P/V/W-NTD around 100 kDa (upper band)
- for V/W NTD, the staining with α -P/V/W-NTD around 55 kDa,
- for W CTD, the staining with α -W-CTD around 55 kDa.

Note that the anti-NTD staining of V and W around 55 kDa (or W NTD in the rNiVΔWCTD condition) cannot be distinguished because of exhibiting roughly the same apparent molecular weight

5. Line 7 page 4 – the idea that W/NTD has ‘no effect’ is not entirely clear to me – it appears to not differ significantly from the control, but also may not differ from WT.

A residual effect of W-NTD cannot be excluded but was not further investigated in this study (see answer 26). The sentence was modified in the text to be less assertive: “In contrast, the expression of W-NTD which is shared with both P and V proteins (Fig. 1a) did not induce a significant effect on the modulation of the NF- κ B activation pathway (Fig. 2a)”.

6. Page 4: “Altogether, these results suggest that NF- κ B inhibitory effect of W occurs downstream IKK β , but cannot counteract the overexpression of p65.”; also pg. 7 “Furthermore, this could explain the inability of W to inhibit the transcriptional activation by overexpressed p65 (Fig. 2d).” – can this be expanded on to clarify why this is expected, particularly if W acts at the level of 14-3-3 interaction with p65.

According to the proposed model of the W protein-mediated inhibition of NF- κ B pathway, the action of W is placed upstream to p65, so it cannot counteract the activity of overexpressed p65. Additional clarifications were added into the revised manuscript (P4, line 18 and P8, line 2).

8. Use of ‘quoted’ – this should probably be ‘indicated’ or ‘denoted’ depending on the context.

Modifications were done in the text and figure legends.

9. Fig 2f – the statistics should probably be used to show whether the NLS0 version of W differs from the WT (this is the key data?)

As explained in the answer to the question 5 above, the figure 2f has been removed from the revised manuscript.

10. Ln36, P4 “W interacts with endogenous 14-3-3 via its CTD S449 residue in” should be “dependent on”?

Modifications were done.

11. Ln9, P4 – somehow should perhaps be somewhat

Modifications were done.

12. Some of the data in the extended figures does not appear to match the text – e.g. “In contrast, the expression of either W-S449A, W-S199A-S449A or W NTD did not affect the IL-1 β -induced p65 intranuclear translocation (Fig. 4c, Extended Data Fig. 3)”.

Corrections were done.

13. “14-3-3 was distributed equally in the cytoplasm and nucleus of the cell (Fig. 5a)” – it appears more nuclear than cytoplasmic to me

Figure 5a was modified with enlarged images from representative individual cells allowing to better appreciate the intracellular localization of 14-3-3 and the included statistical analysis, as mentioned above.

14. Page 6 – references to Extended Data appear incorrect

Modifications were done.

15. Pg 7: “14-3-3 can force the nuclear export of the ligand by enabling the function of its NLS” – is this correct?

The sentence was corrected into: “14-3-3 can force the nuclear export of the ligand by enabling the function of its **NES**”.

16. In discussion, a brief mention of the mechanism of NF-κB deactivation/export with IκBa would be valuable.

Details about the mechanism are mentioned in the third paragraph of the discussion.

17. (Rodriguez, Cruz and Horvath 15 2004) reference in methods should be formatted

The reference has been formatted.

18. In Methods, Analysis of the results from reporter gene assay results could be clearer – e.g. is the normality tested using non-averaged technical replicates across several assays; do all assays compare multiple biological replicates etc. Some areas of the text are not wholly clear.

More details were brought to the Method section for a better clarity (P14, lines 21-36).

19. “RTqPCR analysis of cytokines production” should be bold (title)

Modifications were done.

20. Figure 2d description in legend is not clear.

Additional clarifications were introduced in the legend of the figure 2d.

21. Figure 3b – can a control for specificity of IP (e.g. probe for GAPDH) be included to confirm detection of W in IP from infected cells is specific?

Specificity control using probes for GAPDH were added in the figure 3b, presented below:

22. Although the IP input for HPMEC supports the findings, can a clearer input for HeLa be included?

Following the acquisition of GAPDH probes, we tried to improve the input for HeLa cells, but the results were not much more satisfactory, so we did not include them in the new figure.

23. Fig 5 – why were treated (IL1b) and untreated cells analysed – is there an effect of treatment, and can non-treated cell images be shown?

Both experiments (with or without IL-1 β stimulation) were performed in parallel to analyze a potential effect of IL-1 β treatment. As the effect of W protein was similar in the presence and absence of treatment, only images with treated cells were presented. However, the statistical analysis of the images with nontreated cells is presented in the Supplementary figure 8 bellow.

Supplementary Fig 8. W induces 14-3-3 nuclear accumulation in non-stimulated cells. HPMEC cells were transfected with plasmids encoding W protein constructs or not (\emptyset) or left non-transfected (n.t.). Cells were fixed, permeabilized, stained with DAPI, a α -Flag mouse antibody and anti-14-3-3 rabbit antiserum and analyzed by confocal microscopy. The fluorescence intensity of nuclear 14-3-3 protein was measured in 9 randomly chosen cells and expressed as fold change of the nuclear fluorescence signal of non-transfected non-stimulated cells. Error bars represent the mean's confidence interval (CI 95%) for 9 cells. Statistical significance was assessed by the nonparametric Kruskal-Wallis test with Dunn's multiple comparisons; * $p < 0.05$, ** $p < 0.01$ and *** $p < 0.001$.

24. Fig 5d – Images for infected cells are out of focus/unclear in the pdf.; the zoom appears to select a particularly bright green area; lines to the zoom are incomplete

Better quality of the figure 5d is included in the manuscript as described in the response to the last comment of the Reviewer 1.

25. Extended Fig 3 – is this the intended figure, and why is p65 cytoplasmic in IL1b-treated nt cells? Why is there no p65 in some cells – is this phospho-p65 (and if so, why is it still excluded from the nucleus in nt cells + IL1b)?

This former extended figure 3 presented the antibody control assay (absence of the cross reaction with another cell component and tests of the secondary antibody to avoid any crossing between filters and emission wavelength). As this figure did not provide any critical information for the study understanding, it was removed from the final manuscript.

26. Extended fig 5; some of the cells appear quite different to expected (e.g. some cells appear to exclude 14-3-3 in W-NTD) – there almost appears to be some colocalization with W-NTD – can the authors comment on this.

Better quality of extended figure 5, (Supplementary fig. 7 in the revised manuscript) showing the individual cells in higher magnification to facilitate the evaluation of the intracellular localization of 14-3-3, has been added into the revised manuscript (see below).

Supplementary Fig 7. W-CTD induces 14-3-3 nuclear accumulation. a,b, HPMEC cells were transfected with plasmids encoding indicated W protein constructs or empty vector (\emptyset) or left non-transfected (n.t.). Cells were stimulated 20h later with 10 ng/ml of IL-1 β for 20 min. Then they were fixed, permeabilized, stained with DAPI, a α -Flag mouse antibody and anti-14-3-3 rabbit antiserum and analyzed by confocal microscopy. White arrows show anti-W labelled cells (scale bar = 10 μ m). **b**, The fluorescence intensity of nuclear 14-3-3 protein was measured and expressed as fold change of the nuclear fluorescence signal of non-transfected IL-1 β stimulated cells. Error bars represent the mean's confidence interval (CI 95%) for 9 cells. Statistical significance was assessed by the nonparametric Kruskal-Wallis test with Dunn's multiple comparisons; *p < 0.05.

27. Extended table – requires some revision – it is not clear what +/- indicates, and several comments appear incorrect (e.g. Nuclear import of NiV W is dispensable for NF-kB inhibition)

The extended table has been additionally revised, as presented below.

Supplementary table 2. Summary of available experimental data on the ability of NiV W and its variants to accumulate in the nucleus, to bind to 14-3-3, to alter NF-κB p65 nucleo-cytoplasmic distribution and inhibit NF-κB -mediated signalling

Protein variant & source	NiV W intracellular Distribution	14-3-3		p65 (activated by IL1β/TNFα)			NF-κB Luc activated by IL1β/TNFα	NF-κB activation of transcription by IL1β/TNFα	Comments
		Binds to W	Distribution	Binding to W	Distribution	S536 ^P			
Empty vector						++	++	++	
W		++		-		-	-	-	
P-NTD		-		(-) ³		++	++	++	W-NTD is dispensable
W-CTD		++		(-) ³		-	-	-	W-CTD is required
W-NLS ⁰ 1		(+) ⁴		(-) ³	nt ⁷	nt	++	nt	Nuclear import of NiV W is required for NF-κB inhibition
W-NES ⁰ 2		(+) ⁵		(-) ³	nt	nt	-	nt	NES is dispensable for NiV W mediated inhibition of NF-κB signalling
W-NES ⁰ NLS ⁰		(+) ⁶		(-) ³	nt	nt	++	nt	Loss of competition by importin α3 and α4 => binding of 14-3-3 can occur also in the cytosol
W-S199A		++		(-) ³		-	-	-	
W-S449A		-		(-) ³		++	++	++	W-S449 mediates interaction with 14-3-3 and inhibition of NF-κB signalling
W-S199A-S449A		-		(-) ³		++	++	++	

¹ Mutated aa: K439A - K440A - R442A

² Mutated aa: L174A - L186A

³ Inferred from lack of binding of wt W

⁴ Inferred from W-CTD (lacking NTD in which NES is located)

⁵ Data from Edwards *et al* 2020⁴¹

⁶ Inferred from W-CTD (lacking NTD in which NES is located) and from NLS⁰ data in Edwards *et al* 2020⁴¹

⁷ nt: not tested.

Reviewers' comments:

Reviewer #1 (Remarks to the Author):

The authors have done a good job in addressing comments in what is a fairly dense but thorough description of their project. No additional comments.

Reviewer #2 (Remarks to the Author):

While the paper by Enchery et al. is an interesting study, it remains difficult to follow in places due to fragmentary nature of the data presentation. Furthermore, controls, reproduction and details remain difficult to discern in the body of the paper meaning that full understanding of the paper requires significant investment and this often does not reveal the key information. While I think the paper and findings have significant merit, the current draft submitted appears to require additional work to generate a manuscript easily understandable to the reviewer, including revision of the text, figs/legends, references from the text to figs/legends, ensuring information such as numbers of assay repeats are clear, consistent nomenclature to avoid confusion between content of different figures. The editing does not appear to have included major proof reading as there remain many errors, and as a result, the data has the risk of being mis-communicated. I will include some examples, but cannot cover all errors due to limitations for time.

While there is some additional data in response to queries, analysis appears to be incomplete (e.g. new imaging lacks quantitation), and controls such as analysis of expression of proteins in assays are still not clearly defined/provided.

The authors have provided information to address comments from the previous review, but many points are not clearly covered, and I would ask for additional information/consideration for some of these (detailed below); some additional comments on reading the revised ms are also included:

1. The use of the inhibitory peptide R18 to address dependence of 14-3-3 is a good approach given limitations to knockdown expts. due to multiple isoforms. The data, however, are limited – why is this not subjected to quantitation as has now been used for all other similar assays? Also, in this and other figures, while the materials and methods may specify how many assays were done, reproducibility should be easier to discern from the fig legend or text. I also noted that W appears to cause nuclear localization of R18 – is this consistent?

2. The authors did not provide a response for “Based on the model in Fig. 6, I also wondered whether one might predict greater interaction of 14-3-3 with p65 following wt but not defective mutant W expression.”

3. The authors have responded in part to query 2, but have not clearly addressed the question: “how the proposed model accounts for effects on phosphorylation of p65 (does this fit current knowledge on 14-3-3 and NF-kB)”. Can additional discussion be provided on how the model would account for effects on phosphorylation, and what is known regarding 14-3-3-NFkB interaction and phosphorylation.

4. My original comment: “Extended data fig 1 is cited (In 5 page 4) as showing viability/transfection efficiency for all conditions – I don’t think it does (e.g. different variants, non-treated/treated etc. are missing?).” I think the revised manuscript still implies all conditions in Fig 2 are confirmed, and they are not – only the conditions for fig 2b,c. Variants and conditions for other panels (e.g. CTD, NTD etc) are not confirmed for similar expression or viability. The statement added to the revised ms should be clarified.

5. For point 4, the authors have added discussion of renilla control, which is good. However, it is not clear to me that the westerns in Fig 3 are stated in the manuscript as providing evidence of comparable expression etc. for the assays in Fig 2, as implied in the rebuttal. I also am not convinced that they do - I assume the westerns in Fig 3 are not luciferase assays (and so are transfected differently), do not have equivalent treatment conditions etc. I also do not think that all mutants tested in Fig 2 are shown in westerns in fig 3 (e.g. NES0, NLS0?). To discern the precise relationship of Fig 3 and Fig 2 assays requires extensive cross-checking of the manuscript. As it stands, it is not

easy to discern whether the effects observed in several assays, particularly luciferase assays, relate to expression.

6. Following on from 4, re-assessment of Fig 3 to better understand the data indicated that the HPMEC blot (b) appears poorly aligned so that bands are not clear; furthermore, the lack of detection of W protein WdelCTD IPs does not provide evidence of a lack of interaction (as suggested in the text) since the antibody used appears to be against the CTD, which is absent from this virus. The lack of a control IP is problematic and should be discussed.

7. My original comment: "Figure 2 and other figures testing multiple conditions/W variants for signaling – are there any data for protein expression (e.g. western blot of lysates for viral proteins and loading control);". The response in the rebuttal refers to point 4 – as I state above, I do not think that Fig 3 covers all of the variants tested in Fig 2, or the conditions.

I also would suggest that the authors adopt standard nomenclature for proteins/plasmids, so it is clear where the same or different constructs are used - this was not clear between these figures.

8. The authors have included quantitation for some of the confocal images (ideally all assays including new data should have similar analysis for the reasons outlined in the original review) – all of these appear to use 9 cells, and be from 1 assay for each fig. This appears very limited. Were the assays repeated - can reproducibility be confirmed for specific assays?

9. Original comment: "Suggest short hand use of 'W' could be modified to 'W protein'": There are many instances where this has not been done.

10. Proof reading – it appears perhaps specific instances I identified may have been modified, but there are many places where text remains unclear (see below for some examples); a thorough check is required.

11. Original comment "Figure 1 western – can position of all isoforms be labelled; can expression level of isoforms be estimated by densitometry?" Individual isoforms are not indicated as far as I can see (P, V and W). Is the densitometry included in the rebuttal used in the paper?

12. Original comment: "Some of the data in the extended figures does not appear to match the text – e.g. "In contrast, the expression of either W-S449A, W-S199A-S449A or W NTD did not affect the IL-1b-induced p65 intranuclear translocation (Fig. 4c, Extended Data Fig. 3)". Response: Corrections were done. Ideally the corrections could be specified, as it is unclear what was corrected; the basic meaning of the sentence appears unchanged and still does not appear to match the data in the Supplemental Fig. Similar comment for: "In contrast, 14-3-3 did not accumulate in the nucleus upon expression of W-NTD, W-S449A or W-S199A-S449A variants, although the latter two did accumulate in the nucleus (Fig. 5a, Supplementary Fig. 2 and 7)." Are these the correct figures?

I have not rechecked for other instances of mis-referencing of supplemental data, but this should be done.

13. Title to Fig 3 probably should be revised – is interaction 'via its CTD-S449' ?

14. Fig 5 – what is 'control' in the graph? Title also may be misinterpreted to suggest that WS449 was tested in infected cells for production of cytokines.

15. Importin and KPNA should be defined and then used consistently (either importin or karyopherin/KPN)

16. Textual errors/typos etc. remain common: e.g. The NF- κ B pathway is triggered by multiple sensors of the innate immunity, including...; addition of guanosine residues at an edition site of the P; The sequence of W protein is highly preserved; That loss of NF- κ B inhibitory function correlated with the reduction in the accumulated p65 phosphorylated at Ser536 (phospho-p65). This is not exhaustive.

17. Title: W interaction with endogenous 14-3-3 is dependent on its CTD S449 residue in both transfected and NiV35 infected cells. This may be misinterpreted – is S449 examined in infected cells?

18. "While in both wt NiV- and rNiV-infected cells, W was retrieved in the eluate of immunoprecipitated endogenous 14-3-3, this was not the case after the infection with rNiV W Δ CTD, confirming thus the..." – As discussed above, I think this statement is not correct – WdelCTD cannot be detected by the antibody used, so conclusions on retrieval by IP cannot be made.

19. There are many places where the formatting of figures is difficult to read, or misaligned, text is lost behind panels etc. and so figs should be carefully checked. Legends should also be checked – for

instance, there are no white arrows in Fig. 7.

20. Supplemental Fig 8. Should images be shown?

Rebuttal

We thank the reviewers for their positive appreciations and the useful comments aiming at improving our manuscript.

The manuscript figures and supplemental material has been modified according to reviewer 2 comments.

This include in particular shifting the R18 data as main **Figure 5**, adding data in supplemental **Fig 1b, Fig 2bc**. We took this opportunity to simplify our model (**now Figure 7**) by omitting the NLS0 and S499A variants from the scheme since they are irrelevant to wt NiV infected cells.

Responses in details to reviewer 2 comments:

Reviewer 2

While the paper by Enchery et al. is an interesting study, it remains difficult to follow in places due to fragmentary nature of the data presentation. Furthermore, controls, reproduction and details remain difficult to discern in the body of the paper meaning that full understanding of the paper requires significant investment and this often does not reveal the key information. While I think the paper and findings have significant merit, the current draft submitted appears to require additional work to generate a manuscript easily understandable to the reviewer, including revision of the text, figs/legends, references from the text to figs/legends, ensuring information such as numbers of assay repeats are clear, consistent nomenclature to avoid confusion between content of different figures.

The editing does not appear to have included major proof reading as there remain many errors, and as a result, the data has the risk of being mis-communicated. I will include some examples, but cannot cover all errors due to limitations for time.

While there is some additional data in response to queries, analysis appears to be incomplete (e.g. new imaging lacks quantitation), and controls such as analysis of expression of proteins in assays are still not clearly defined/provided.

The authors have provided information to address comments from the previous review, but many points are not clearly covered, and I would ask for additional information/consideration for some of these (detailed below); some additional comments on reading the revised ms are also included:

1. The use of the inhibitory peptide R18 to address dependence of 14-3-3 is a good approach given limitations to knockdown expts. due to multiple isoforms. The data, however, are limited – why is this not subjected to quantitation as has now been used for all other similar assays? Also, in this and other figures, while the materials and methods may specify how many assays were done, reproducibility should be easier to discern from the fig legend or text. I also noted that W appears to cause nuclear localization of R18 – is this consistent?

Response: The quantitation for R18 transfected cells has been done and now shown as new Figure 5b. The same protocol has been performed for all imaging experiments and quantifications presented in the figures and the number of assays has been specified. It is correct that the image that was shown in supplementary Figure 6 corresponding to W transfected cells possibly showed nuclear localization of R18. After reanalyzing all images this turned out to be an exception. Hence we have substituted the image set for W transfected cells by more representative one to avoid confusion. As R18 data are important for the conclusion of the manuscript, we have moved the Supplementary Figure 6 into the principal Figure 5.

2. The authors did not provide a response for “Based on the model in Fig. 6, I also wondered whether one might predict greater interaction of 14-3-3 with p65 following wt but not defective mutant W expression.”

Response: We apologize for not having discuss this interesting hypothesis. 14.3.3 forms homo- and hetero-dimers and this explains it can bind simultaneously two different ligands. Then, one can imagine the binding of a first ligand to one monomer can enhance by allostery the binding to a second ligand to the other

monomer. To the best of our knowledge this has not been described for 14.3.3 so far. Instead, monomers binds to their ligand as well as dimers (see <http://dx.doi.org/10.1016/j.febslet.2012.10.048> for review), hence we prefer to avoid mentioning this hypothesis that looks over-speculative. An alternative could have been that W also binds to p65 with a weak affinity that could favor the formation of W/14-3-3/p65 complexes, but this would not fit with our observation of a reduced co-ip of 14-3-3 upon overexpression of p65.

3. The authors have responded in part to query 2, but have not clearly addressed the question: “how the proposed model accounts for effects on phosphorylation of p65 (does this fit current knowledge on 14-3-3 and NF-κB)”. Can additional discussion be provided on how the model would account for effects on phosphorylation, and what is known regarding 14-3-3-NFκB interaction and phosphorylation.

Response: We apologize for having misunderstood the query. Our proposed model fits the current knowledge regarding what is already described concerning 14-3-3/p65 interactions and in the presence or not of NiV W protein. This is now discussed in more detail lines 318-329: “According to this model, the W protein could reinforce the physiological repressing effect of 14-3-3 on NF-κB transcriptional activity by forcing nuclear accumulation of 14-3-3 in agreement with 14-3-3 ensuring the efficient nuclear export of NF-κB p65/p50 complexes¹⁰ and the pleiotropic inhibitory effect the W protein on the cellular gene transcription recently reported⁴¹. Globally, our proposed mechanism fits with the current knowledge regarding to the interaction of 14-3-3 with p65 regulated by phosphorylation of the later. Although the W protein does not interact with p65 and seems rather to compete out with p65 for binding to 14.3.3 within the nucleus, we cannot exclude that W protein can alter the phosphorylation profile of p65. Indeed, the nuclear trafficking of p65 is finely tuned by successive phosphorylation steps: phosphorylation on the residue S536 of p65 enables its nuclear localization (and transactivation activity)⁵, then, it is exported outside the nucleus p65 upon its phosphorylated at positions S42, S281 and S340 that allows its association with 14-3-3 and IκBα as cargo export machinery¹⁰.”

4. My original comment: “Extended data fig 1 is cited (In 5 page 4) as showing viability/transfection efficiency for all conditions – I don’t think it does (e.g. different variants, non-treated/treated etc. are missing?).” I think the revised manuscript still implies all conditions in Fig 2 are confirmed, and they are not – only the conditions for fig 2b,c. Variants and conditions for other panels (e.g. CTD, NTD etc) are not confirmed for similar expression or viability. The statement added to the revised ms should be clarified.

Response: It is true, it is an oversight. The statement (now mentioning Extended data fig 2 in the new version) has been corrected.

5. For point 4, the authors have added discussion of renilla control, which is good. However, it is not clear to me that the westerns in Fig 3 are stated in the manuscript as providing evidence of comparable expression etc. for the assays in Fig 2, as implied in the rebuttal. I also am not convinced that they do - I assume the westerns in Fig 3 are not luciferase assays (and so are transfected differently), do not have equivalent treatment conditions etc. I also do not think that all mutants tested in Fig 2 are shown in westerns in fig 3 (e.g. NES0, NLS0?). To discern the precise relationship of Fig 3 and Fig 2 assays requires extensive cross-checking of the manuscript. As it stands, it is not easy to discern whether the effects observed in several assays, particularly luciferase assays, relate to expression.

Response: Indeed, conditions of westerns in Fig. 3 were not exactly the same as the ones in Fig. 2, and westerns in Fig. 3 are not linked to luciferase assays. To illustrate that FLAG-tagged W protein constructs were roughly expressed at similar level in a given cell, their expression levels in HeLa cells as detected by western blot using anti-FLAG antibody are now shown in Supplemental Fig 2b,c. We are thus confident that the functional differences between the W protein constructs cannot be explained by a major difference in their expression levels.

6. Following on from 4, re-assessment of Fig 3 to better understand the data indicated that the HPMEC blot (b) appears poorly aligned so that bands are not clear; furthermore, the lack of detection of W protein WΔCTD IPs does not provide evidence of a lack of interaction (as suggested in the text) since the antibody used appears to be against the CTD, which is absent from this virus. The lack of a control IP is problematic and should be discussed.

Response: The HPMEC blot corresponding to the anti-W-CTD detection has been realigned.

The obvious lack of detection of W protein in the rNiV-WΔCTD by using an anti-W-CTD, is now pointed out lines 195-194 We did not have any other choice to fish only W protein since none of the proteins in the recombinant viruses are tagged.

7. My original comment: "Figure 2 and other figures testing multiple conditions/W variants for signaling – are there any data for protein expression (e.g. western blot of lysates for viral proteins and loading control);". The response in the rebuttal refers to point 4 – as I state above, I do not think that Fig 3 covers all of the variants tested in Fig 2, or the conditions.

I also would suggest that the authors adopt standard nomenclature for proteins/plasmids, so it is clear where the same or different constructs are used - this was not clear between these figures.

Response: Western blot showing the relative expression of all W protein variant has been added in Fig 2b,c (see our answer to point 5 above)

The nomenclature of W protein variants has been homogenized throughout the ms. In addition all figures have been now clearly labelled with the type of protein construct and antibody used to detect them (mostly anti-FLAG since all W protein variants are tagged at their N-terminus).

8. *The authors have included quantitation for some of the confocal images (ideally all assays including new data should have similar analysis for the reasons outlined in the original review) – all of these appear to use 9 cells, and be from 1 assay for each fig. This appears very limited. Were the assays repeated - can reproducibility be confirmed for specific assays?*

Response: With the exception of Supplementary Figure 8) every nuclear imaging assay now include quantitation. Each experiment has been done two to three times usually in triplicate or duplicate (as stated accordingly) and 9 cells taken from 9 different fields were selected randomly by the device or by the user. Shown quantitation of nuclear localization of 14-3-3 or p65 are from a single representative analysis of randomly selected pictures of cells taken within the same confocal analysis session to minimize any possible technical induced variation between imaging sessions. This point has been added in the Methods section lines 648-652.

9. *Original comment: “Suggest short hand use of ‘W’ could be modified to ‘W protein’”: There are many instances where this has not been done.*

We apologize. W has been modified in W protein in the entire text, except then protein has already been mentioned in the same sentence or than W is followed by the name of the mutant (W-CTD, W-S449A etc) to avoid repetitions and too long sentences.

10. *Proof reading – it appears perhaps specific instances I identified may have been modified, but there are many places where text remains unclear (see below for some examples); a thorough check is required.*

The extensive proof reading of the entire manuscript has been done by native English speaking virologist.

11. *Original comment “Figure 1 western – can position of all isoforms be labelled; can expression level of isoforms be estimated by densitometry?” Individual isoforms are not indicated as far as I can see (P, V and W). Is the densitometry included in the rebuttal used in the paper?*

Response: Figure 1 has been completed with clear indication of the identity of each band, as well as the caption. The densitometry results are now part of supplementary Fig. 1b .

12. *Original comment: “Some of the data in the extended figures does not appear to match the text – e.g. “In contrast, the expression of either W-S449A, W-S199A-S449A or W NTD did not affect the IL-1b-induced p65 intranuclear translocation (Fig. 4c, Extended Data Fig. 3)”. Response: Corrections were done. Ideally the corrections could be specified, as it is unclear what was corrected; the basic meaning of the sentence appears unchanged and still does not appear to match the data in the Supplemental Fig. Similar comment for: “In contrast, 14-3-3 did not accumulate in the nucleus upon expression of W-NTD, W-S449A or W-S199A-S449A variants, although the latter two did accumulate in the nucleus (Fig. 5a, Supplementary Fig. 2 and 7).” Are these the correct figures?*

I have not rechecked for other instances of mis-referencing of supplemental data, but this should be done.

The first sentence have been modified into: “In contrast, the expression of W-S449A, W-S199A-S449A or W-NTD proteins did not affect the IL-1-induced p65 intranuclear translocation (Fig. 4c,d), lines 224_226

The second sentence has been changed as well with correct figure referencing. The first paragraph of the section now entitled “The W protein induces nuclear accumulation of 14-3-3 both *in vitro* and *in vivo* and inhibits the production of pro-inflammatory cytokines “ has been reshuffled in order to logically start with the resting condition, i.e. without stimulation by IL-1beta (now supplementary Figure 6), lines 235-248.

The referencing of the supplementary data has been carefully checked elsewhere.

13. Title to Fig 3 probably should be revised – is interaction ‘via its CTD-S449’ ?

Title of the Fig 3 has been modified into: “W protein requires CTD-S449 to interact with 14-3-3...”

14. Fig 5 – what is ‘control’ in the graph? Title also may be misinterpreted to suggest that WS449 was tested in infected cells for production of cytokines.

Response: The “control” category is now clarified in the legend of Figure 6d (renumbered Figure 5d) and in the Fig. 6d itself as shown below.

In this experiment, we performed immunofluorescence assay on lung section obtained from NiV-infected African green monkeys (AGM) and from non-infected AGM, using anti-W-CTD polyclonal Ab and anti-14-3-3 pan Ab on each condition. For the n.i. condition, we selected cells with no discriminatory condition on non-infected slices. For the NiV-infected AGM condition, we selected lung cells that showed a NiV-W-CTD staining, i.e. infected, and, then, we measured the 14-3-3 staining in the nucleus. As an internal control, we used the 14-3-3 staining level in the nucleus of selected cells within the same field that did not show any staining with the anti-NiV-WCTD and thus considered as being non-infected. The title of Figure 5 (now Figure 6) has been additionally clarified.

“**Figure 6.** W induces 14-3-3 nuclear accumulation after transient expression *in vitro* and *in vivo* in lung cells in NiV-infected monkeys and suppresses the production of proinflammatory cytokines *in vitro*”

15. *Importin and KPNA should be defined and then used consistently (either importin or karyopherin/KPN)*

Importin has been defined in the introduction, lines 57-58, and is now used consistently throughout the revised manuscript.

16. *Textual errors/typos etc. remain common: e.g. The NF- κ B pathway is triggered by multiple sensors of the innate immunity, including...; addition of guanosine residues at an edition site of the P; The sequence of W protein is highly preserved; That loss of NF- κ B inhibitory function correlated with the reduction in the accumulated p65 phosphorylated at Ser536 (phospho-p65). This is not exhaustive.*

These sentences have been corrected and the manuscript has been thoroughly edited. We hope that this will be satisfactory.

17. *Title: W interaction with endogenous 14-3-3 is dependent on its CTD S449 residue in both transfected and NiV35 infected cells. This may be misinterpreted – is S449 examined in infected cells?*

Due to the limited availability of NiV recombinant viruses only W-CTD was analyzed in NiV-infected cells, and the importance of CTD S449 was deduced from the transfections studies. To avoid any misinterpretation, we have modified the title into “W protein interaction with endogenous 14-3-3 is dependent on W-CTD in both transfected and NiV-infected cells and the interaction relies on S449 residue.” Lines 168-169.

18. *“While in both wt NiV- and rNiV-infected cells, W was retrieved in the eluate of immunoprecipitated endogenous 14-3-3, this was not the case after the infection with rNiV W Δ CTD, confirming thus the...” – As discussed above, I think this statement is not correct – W Δ CTD cannot be detected by the antibody used, so conclusions on retrieval by IP cannot be made.*

Response: Indeed, W Δ CTD cannot be detected by an antibody that binds the CTD of W. The text has been modified accordingly (lines 193-194).

19. *There are many places where the formatting of figures is difficult to read, or misaligned, text is lost behind panels etc. and so figs should be carefully checked. Legends should also be checked – for instance, there are no white arrows in Fig. 7.*

Response: As done for the text, all figures have been reformatted and their legends carefully edited (see for example figure parts shown above). We are sorry for having left “white arrows” that were mistakenly left from a preliminary version of Figure 7.

20. *Supplemental Fig 8. Should images be shown?*

The full set of data (images and quantification) obtained without and with IL-1-beta stimulation is now shown (Supplementary Fig 7 (without IL-1) and Figure 6 and supplementary Fig 8 (with IL-1))

REVIEWERS' COMMENTS:

Reviewer #2 (Remarks to the Author):

The ms. is much improved. I have only minor suggestions:

1. pg 4 ln 139 - check names of constructs mW-CTD, W-CTDm
2. pg 6 Please consider rewording lines 256-257; same for the title of Fig 6 - while the interpretation overall is consistent with the combined data in the paper, I think the wording can still be misinterpreted that the W protein was specifically shown to have this role in vivo by the data, and I don't think it was.
3. Fig 2b - should labelling be included to indicate the type of FLAG transfection?
4. Figure 4&5. It is still not 100% clear how data derived - is it 9 cells from 1 assay typical of several assays, or 9 cells combined from several assays (e.g. 3 cells from each of 3 assays)?
5. Figure 6 - specify number of assays.
6. Sup Fig 2 and other figs - check that type of error bars, n values etc to derive errors are included.

Rebuttal letter

We thank the reviewer for the useful comments aiming at improving our manuscript.

The manuscript figures and supplemental material have been modified accordingly to reviewer's comments, as described below.

1. pg 4 ln 139 - check names of constructs mW-CTD, W-CTDm

The names of the constructs have been corrected in the revised manuscript accordingly to reviewer's pertinent remark.

2. pg 6 Please consider rewording lines 256-257; same for the title of Fig 6 - while the interpretation overall is consistent with the combined data in the paper, I think the wording can still be misinterpreted that the W protein was specifically shown to have this role in vivo by the data, and I don't think it was.

According to the reviewer's suggestions, the indicated lines as well as the title of the figure 6 have been modified.

3. Fig 2b - should labelling be included to indicate the type of FLAG transfection?

We thank the reviewer for its remark and we rectified the observed discrepancy. The images initially chosen to present our results were indeed confusing regarding the condition for FLAG staining. Indeed, in the images presented originally in the figure 2b originated from different experimental sets, where in one of the sets, an empty plasmid that did not code for FLAG was used in the experiment presented in the figure 2b to better discriminate the "m" from the "W" condition. However, as noticed by the reviewer, the description of the experiment in the methods and figure legend did not match those images. In order to avoid any confusion and remain coherent with the results presented in 2b and 2c, we replaced all images from the figure 2b with images representative where the same experimental procedure was used and all plasmid constructions coded for FLAG that corresponded also to the experiments used to perpetrate results presented in the figure 2c.

4. Figure 4&5. It is still not 100% clear how data derived - is it 9 cells from 1 assay typical of several assays, or 9 cells combined from several assays (e.g. 3 cells from each of 3 assays)? During the analysis of the obtained images from 3 independent experiments, a total of 9 cells combined from those 3 assays were analyzed. The additional explanation has been added into the legends of the figure 4 and 5.

5. Figure 6 - specify number of assays.

The number of assays has been specified in the legend of Figure 6.

6. Sup Fig 2 and other figs - check that type of error bars, n values etc to derive errors are included.

Error bars and n values have been included in the supplementary figure 2 and in other figure legends where it was requested.

Finally, the re-examination of the raw data with full blots, all included in the supplementary data set of the revised manuscript, allowed us to correct few errors in the figure 3b and c, with lines presenting certain bands in gels, which we describe below in detail. None of those discrepancies had any impact in the analysis and the conclusions associated to those figures in the reviewed manuscript.

Figure 3B: Western blot analysis

We would like to inform you about the modification that has been performed in the presented western blot images. By compiling the raw data associated to that specific figure, we realized that in the HPMEC condition (gels on the right side), the immuno-blots representing 14-3-3 were inverted between “IP 14-3-3” and “Input” samples (highlighted gels in red), which has been now corrected in the revised manuscript.

Figure 3C: Western blot analysis

By compiling the raw data, we realized that gels presenting GAPDH in the “Input” samples from both HeLa and HPMEC conditions did not correspond to correct bands. This has been corrected in the revised manuscript and modified lines are highlighted gels in red below.